# Improving Entropic Out-of-Distribution Detection using Isometric Distances and the Minimum Distance Score

## Abstract

Current out-of-distribution detection approaches usually present special require-
ments (e.g., collecting outlier data and hyperparameter validation) and produce
side effects (classification accuracy drop and slow/inefficient inferences). Recently,
entropic out-of-distribution detection has been proposed as a seamless approach
(i.e., a solution that avoids all the previously mentioned drawbacks). The entropic
out-of-distribution detection solution comprises the IsoMax loss for training and
the entropic score for out-of-distribution detection. The IsoMax loss works as a
SoftMax loss drop-in replacement because swapping the SoftMax loss with the
IsoMax loss requires no changes in the model's architecture or training proce-
dures/hyperparameters. In this paper, we propose to perform what we call an
isometrization of the distances used in the IsoMax loss. Additionally, we propose
to replace the entropic score with the minimum distance score. Our experiments
showed that these simple modifications increase out-of-distribution detection per-
formance while keeping the solution seamless.

## 1 Introduction

Neural networks have been used in classification tasks in many real-world applications [4]. In such
cases, the system usually needs to be able to identify whether a given input belongs to any of the
classes on which it was trained. Hendrycks & Gimpel [9] called this capability out-of-distribution
(OOD) detection and proposed datasets and metrics to allow standardized performance evaluation
and comparison. However, current OOD detection solutions still present limitations (e.g., special
requirements and side effects) that prevent a more general use of OOD detection capabilities in
practical real-world applications [27] (Table 1).

First, OOD detection solutions commonly present hyperparameters that usually presume access to
out-of-distribution samples to be defined [23, 22, 19, 18, 3]. A consequence of presuming access to
OOD samples to validate hyperparameters and using the same distribution to evaluate OOD detection
results is producing overestimated performance estimations [32]. To avoid unrealistic access to OOD
samples and overestimated performance, Lee et al. [19] proposed to validate hyperparameters using
adversarial samples. However, this requires the generation of adversarial examples. Moreover, this
procedure requires the determination of hyperparameters (e.g., maximum adversarial perturbation)
typically unknown when dealing with novel datasets. Similar arguments hold for solutions based on
adversarial training [8, 17, 21, 14, 18], which also result in higher training time. Approaches based
on the generation of adversarial examples or the use of adversarial training may also have limited
scalability when dealing with large images such as those presented in the ImageNet [2].

Table 1: Out-of-distribution detection approaches: special requirements and side effects.

| Approach | Special Requirement | | Side Effect | |
|---|---|---|---|---|
| | Hyperparameter Tuning | Outlier Data | Slow/Inefficient Inference | Classification Accuracy Drop |
| ODIN [23] | Required | Not Required | Present | Not Present |
| Mahalanobis [19] | Required | Not Required | Present | Not Present |
| ACET [8] | Required | Not Required | Not Present | Present |
| Outlier Exposure [10] | Not Required | Required | Not Present | Not Present |
| Generalized ODIN [11] | Required | Not Required | Present | Present |
| Gram Matrices [30] | Not Required | Not Required | Present | Not Present |
| Scaled Cosine [34] | Not Required | Not Required | Not Present | Present |
| Energy-based [25] | Required | Required | Not Present | Not Present |
| **Entropic (Seamless) [27, 26] IsoMax + Entropic Score** | **Not Required** | **Not Required** | **Not Present** | **Not Present** |
| **Entropic (Seamless) [ours] IsoMax$_\mathcal{I}$ + MinDistance Score** | **Not Required** | **Not Required** | **Not Present** | **Not Present** |

Many solutions make use of the so-called *input preprocessing* technique introduced in ODIN [23]. However, the use of the mentioned technique *increases at least four times the inference delay and power consumption* [27] since a combination of a first forward pass, backpropagation operation, and second forward pass is required [23, 19, 11, 3] for a single useful inference. Actually, approaches that may be applied directly to pretrained models and altogether avoid training or fine-tuning the model [23, 19, 30] usually produce inefficient inferences and/or additional computational complexity to perform OOD detection [26, Section IV, D]. *From a practical point of view, this is a drawback, as inferences may be performed thousands or millions of times in the field.* Hence, such approaches may be prohibitive (not sustainable) from environmental [31][1] and real-world cost-based perspectives.

Another harmful common side effect is the so-called *classification accuracy drop*[2] [34, 11]. In such cases, higher OOD detection performance is achieved at the expense of a drop in the classification accuracy compared with models trained using the usual SoftMax loss (i.e., the combination of the SoftMax activation and the cross-entropy loss [24]). From a practical perspective, this situation is undesired because the detection of out-of-distribution samples may be a rare event. At the same time, the classification is the main aim of the designed system [1].

Hsu et al. [11] proposed to use the in-distribution validation set to avoid the need for accessing OOD samples to determine the hyperparameters required by the solution. However, considering that CIFAR10 and CIFAR100 do not have separated sets for validation and testing, the results may also be overestimated because the validation sets used to define the hyperparameters were reused for OOD detection performance estimation. A more realistic OOD detection performance estimation could have been achieved by removing the in-distribution validation set from in-distribution training data. However, this would probably produce an even higher *classification accuracy drop*. Additionally, the solution proposed in [11] is expensive and not environment-friendly, as it uses *input preprocessing* and, consequently, produces slow and energy-inefficient inferences [27, 26]. Recently, many OOD detection approaches have used additional/extra/outlier data [10, 25, 5]. The Gram matrices solution calculates values produced by the model during inference [30] to perform OOD detection.

In some cases, an ensemble of classifiers is used [35]. For deep ensembles, Lakshminarayanan et al. [17] proposed an ensemble of same-architecture models trained with different random initial weights. Some proposals required model structural changes to tackle OOD detection [37], and certain trials used uncertainty or confidence estimation/calibration techniques [13, 20, 28, 16, 33]. However, Bayesian neural networks used in most of these are usually harder to implement and require

---

[1] `https://www.youtube.com/watch?v=KnOpWgUCtaM`

[2] In this paper, we consider that an approach does not present classification accuracy drop if it always presents a classification accuracy higher or less than one percent (1%) lower than SoftMax-loss-trained models.

much more computational resorces to train. Moreover, computational constraints usually require approximations that compromise the performance, which is also affected by the prior distribution used [17]. For example, MC-dropout uses pretrained models with dropout activated during the test time. An average of many inferences is used to perform a single decision [6].

The entropic out-of-distribution detection approach, which is composed of the IsoMax loss for training and the entropic score for OOD detection, avoids all mentioned special requirements and side effects [27]. Indeed, no hyperparameter tuning is required because *the entropic scale is a global constant kept equal to ten for all combinations of datasets and models*. Even if we call the entropic scale a "hyperparameter", the IsoMax does not involve hyperparameter *tuning*, as the same constant value of entropic scale is used in all situations. This is possible because Macêdo et al. experimentally showed in [27, Fig. 3] and in [26, Section IV, A] that *the OOD detection performance presents a well-behaved dependence on the entropic scale regardless of the dataset and model*. No additional/extra/outlier data are necessary. Models trained using IsoMax loss produce inferences as fast and energy-efficient as the inferences produced by SoftMax-loss-trained networks. The OOD detection requires only a speedy entropy calculation. Finally, no classification accuracy drop is observed.

**Contributions**  Our contribution in this paper is threefold: First, in addition to minor changes, we perform what we call an isometrization of the *feature-prototype distances* used by the IsoMax loss. We call our modified version of IsoMax the *isometric* isotropy maximization loss or *isometric* IsoMax loss (IsoMax$_\mathcal{I}$ loss). Second, we propose to use the *minimum feature-prototype distance* as the score to perform OOD detection. Considering that the minimum feature-prototype distance is calculated to perform the classification, *the OOD detection task presents essentially zero computational cost* because we simply reuse this value as the score to perform OOD detection. Third, in addition to experimental evidence, we provide insights into why a combination of training using the *isometric distances* provided by IsoMax$_\mathcal{I}$ and performing OOD detection using the minimum distance scores produces a substantial performance increase in OOD detection compared to IsoMax combined with the entropic score. *Our approach keeps the solution seamless (i.e., it avoids the previously mentioned special requirements and side effects) while significantly increasing the OOD detection performance.* Similar to IsoMax loss, IsoMax$_\mathcal{I}$ works as a SoftMax loss drop-in replacement, as no procedures other than regular neural network training are required.

## 2   Isometric Distances and Minimum Distance Score

**Isometric Distances**  Consider an input $\boldsymbol{x}$ applied to a neural network that performs a parametrized transformation $\boldsymbol{f_\theta(x)}$. Moreover, consider $\boldsymbol{p}_\phi^j$ be the learnable prototype associated with the class $j$. Additionally, let the expression $\|\boldsymbol{f_\theta(x)} - \boldsymbol{p}_\phi^j\|$ represent the *nonsquared* Euclidean distance between $\boldsymbol{f_\theta(x)}$ and $\boldsymbol{p}_\phi^j$. Finally, consider $\boldsymbol{p}_\phi^k$ as a learnable prototype associated with the correct class for the input $\boldsymbol{x}$. Hence, we write the IsoMax loss [27] for a batch of $N$ examples using the equation below:

$$\mathcal{L}_{\mathsf{IsoMax}} = -\frac{1}{N} \sum_{k=1}^{N} \log \left( \frac{\exp(-E_s \|\boldsymbol{f_\theta(x)} - \boldsymbol{p}_\phi^k\|)}{\sum_j \exp(-E_s \|\boldsymbol{f_\theta(x)} - \boldsymbol{p}_\phi^j\|)} \right) \tag{1}$$

In the above equation, the $E_s$ represents the entropic scale. From Equation (1), we observe that the distances from IsoMax loss are given by the expression $\mathcal{D} = \|\boldsymbol{f_\theta(x)} - \boldsymbol{p}_\phi^j\|$. During inference, probabilities calculated based on these distances are used to produce the negative entropy, which serves as a score to perform OOD detection. However, as the features $\boldsymbol{f_\theta(x)}$ are unnormalized, examples with low norms are unjustifiably favored to be considered OOD examples since they tend to produce high entropy. Additionally, as the weights $\boldsymbol{p}_\phi^j$ are unnormalized, examples from classes that present prototypes with low norms are unjustifiably favored to be considered OOD examples for the same reason.

Hence, we propose to replace $\boldsymbol{f_\theta(x)}$ with its normalized version given by $\widehat{\boldsymbol{f_\theta(x)}} = \boldsymbol{f_\theta(x)}/\|\boldsymbol{f_\theta(x)}\|$.

Additionally, we propose to replace $\boldsymbol{p}_\phi^j$ with its normalized version given by $\widehat{\boldsymbol{p}_\phi^j} = \boldsymbol{p}_\phi^j/\|\boldsymbol{p}_\phi^j\|$. The expression $\|\boldsymbol{v}\|$ represents the 2-norm of a given vector $\boldsymbol{v}$.

Table 2: **Classification accuracy of models trained using SoftMax, IsoMax, and IsoMax$_\mathcal{I}$ losses**. In addition to avoiding classification accuracy drop compared with SoftMax-loss- and IsoMax-loss-trained networks, IsoMax$_\mathcal{I}$-loss-trained models show higher OOD detection performance (Table 3).

| Model | Data | Train Accuracy (%) [↑] | Test Accuracy (%) [↑] |
| --- | --- | --- | --- |
| | | SoftMax Loss / IsoMax Loss / IsoMax$_\mathcal{I}$ Loss | |
| DenseNetBC100 | CIFAR10 | 99.9 / 99.9 / 99.9 | 95.4 / 95.2 / 95.2 |
| | CIFAR100 | 99.9 / 99.0 / 99.9 | 77.5 / 77.5 / 76.8 |
| | SVHN | 96.9 / 97.6 / 97.1 | 96.6 / 96.6 / 96.6 |
| ResNet110 | CIFAR10 | 99.9 / 99.9 / 99.9 | 94.5 / 94.6 / 94.6 |
| | CIFAR100 | 99.5 / 99.9 / 99.8 | 72.7 / 74.1 / 73.9 |
| | SVHN | 99.8 / 99.9 / 99.5 | 96.7 / 96.9 / 96.9 |

However, while the distances in the original IsoMax loss may vary from zero to infinity, the distance between two normalized vectors is always equal to or lower than two. To avoid this unjustifiable and unreasonable restriction, we introduce the *distance scale $d_s$*, which is a *scalar learnable parameter*. Naturally, we require the distance scale to always be positive by taking its absolute value $|d_s|$.

The feature normalization makes the solution isometric regardless of the norm of the features produced by the examples. The distance scale is class independent, as it is a *single* scalar value regularly learnable during training. The weight normalization and the class independence of the distance scale make the solution isometric regarding all classes. Hence, the proposed distance is isometric because it produces an isometric treatment of all features, prototypes, and classes. Therefore, we can write the expression for the *isometric distances* used by the IsoMax$_\mathcal{I}$ loss as:

$$\mathcal{D}_\mathcal{I} = |d_s| \, \|\widehat{\boldsymbol{f_\theta}(\boldsymbol{x})} - \widehat{\boldsymbol{p}_\phi^j}\| \tag{2}$$

Returning to Equation (1), we can write the expression for the IsoMax$_\mathcal{I}$ loss as follows:

$$\mathcal{L}_{\mathsf{IsoMax}_\mathcal{I}} = -\frac{1}{N} \sum_{k=1}^{N} \log \left( \frac{\exp(-E_s \, |d_s| \, \|\widehat{\boldsymbol{f_\theta}(\boldsymbol{x})} - \widehat{\boldsymbol{p}_\phi^k}\|)}{\sum_j \exp(-E_s \, |d_s| \, \|\widehat{\boldsymbol{f_\theta}(\boldsymbol{x})} - \widehat{\boldsymbol{p}_\phi^j}\|)} \right) \tag{3}$$

Applying the entropy maximization trick (i.e., the removal of the entropic score $E_s$ for inference) [27], we can write the expression for the IsoMax$_\mathcal{I}$ loss probabilities used during inference for performing OOD detection when using the entropic score [27]:

$$\mathcal{P}_{\mathsf{IsoMax}_\mathcal{I}}(y^{(i)}|\boldsymbol{x}) = \frac{\exp(-\,|d_s| \, \|\widehat{\boldsymbol{f_\theta}(\boldsymbol{x})} - \widehat{\boldsymbol{p}_\phi^i}\|)}{\sum_j \exp(-\,|d_s| \, \|\widehat{\boldsymbol{f_\theta}(\boldsymbol{x})} - \widehat{\boldsymbol{p}_\phi^j}\|)} \tag{4}$$

Different from IsoMax loss where the prototypes are initialized to a zero vector, we initialized all prototypes using a normal distribution with a mean of zero and standard deviation of one. This approach is necessary because we normalize the prototypes when using IsoMax$_\mathcal{I}$ loss. The distance scale is initialized to one. We add no hyperparameters to the solution.

**Minimum Distance Score**   Motivated by the desired characteristics of the isometric distances used in IsoMax$_\mathcal{I}$, we propose to use what we call the minimum distance as the score for performing OOD detection. Naturally, the minimum distance score for the IsoMax$_\mathcal{I}$ is given by:

$$\mathcal{S}_{\mathsf{MinDistance}} = \min_j \left( \|\widehat{\boldsymbol{f_\theta}(\boldsymbol{x})} - \widehat{\boldsymbol{p}_\phi^j}\| \right) \tag{5}$$

Table 3: **Fair comparison of seamless approaches: No hyperparameter tuning, no additional/extra/outlier data, no classification accuracy drop, and no slow/inefficient inferences.** SoftMax+ES means training using SoftMax loss and performing OOD detection using the entropic score (ES). IsoMax+ES means training using IsoMax loss and performing OOD detection using the entropic score (ES). IsoMax$_\mathcal{I}$+MDS means training using IsoMax$_\mathcal{I}$ loss and performing OOD detection using minimum distance score (MDS). The best results are in bold (0.5% tolerance).

| Model | Data (training) | OOD (unseen) | Out-of-Distribution Detection: Seamless Approaches. | |
|---|---|---|---|---|
| | | | TNR@TPR95 (%) [↑] | AUROC (%) [↑] |
| | | | SoftMax+ES / IsoMax+ES / IsoMax$_\mathcal{I}$+MDS (ours) | |
| DenseNetBC100 | CIFAR10 | SVHN | 33.2 / 77.0 / **97.2** | 86.9 / 96.6 / **99.5** |
| | | TinyImageNet | 59.8 / 88.0 / **92.5** | 94.2 / 97.8 / **98.6** |
| | | LSUN | 69.5 / 94.5 / **95.3** | 95.9 / **98.8** / **99.1** |
| | CIFAR100 | SVHN | 24.9 / 23.4 / **78.6** | 81.9 / 88.6 / **96.5** |
| | | TinyImageNet | 23.7 / 49.1 / **85.6** | 78.8 / 92.6 / **97.6** |
| | | LSUN | 24.4 / 63.0 / **83.4** | 77.9 / 94.7 / **97.4** |
| | SVHN | CIFAR10 | 83.7 / 94.1 / **95.3** | 96.9 / 98.5 / **99.1** |
| | | TinyImageNet | 90.0 / 97.0 / **98.3** | 98.1 / 99.1 / **99.7** |
| | | LSUN | 88.4 / 96.8 / **97.8** | 97.8 / 99.1 / **99.7** |
| ResNet110 | CIFAR10 | SVHN | 37.8 / 73.0 / **83.6** | 89.6 / 95.1 / **97.3** |
| | | TinyImageNet | 43.7 / 73.7 / **75.5** | 90.6 / **95.9** / **96.0** |
| | | LSUN | 52.1 / 82.8 / **86.3** | 92.8 / 96.9 / **97.7** |
| | CIFAR100 | SVHN | 15.4 / 18.7 / **30.7** | 67.5 / 84.7 / **85.8** |
| | | TinyImageNet | 18.8 / 26.3 / **42.9** | 73.5 / 84.5 / **87.9** |
| | | LSUN | 21.3 / 30.2 / **46.9** | 76.4 / 87.1 / **89.4** |
| | SVHN | CIFAR10 | 68.6 / **80.4** / 72.0 | 91.7 / **95.2** / 93.3 |
| | | TinyImageNet | 71.7 / **84.4** / 83.1 | 93.1 / **95.8** / 96.2 |
| | | LSUN | 69.1 / **80.4** / 76.3 | 91.8 / **94.3** / **94.3** |

In the previous equation, $|d_s|$ was removed because it is a scale factor that does not change after the training is completed. The minimum distance is computed to perform the classification, as the predicted class is the one that presents *the lowest feature-prototype distance*. Therefore, when using this score, *the OOD detection presents essentially zero latency and computational cost, as we simply reuse the minimum distance already calculated*.

# 3 Experiments

To allow standardized comparison, we used the datasets, training procedures, and metrics that were established in Hendrycks & Gimpel [9] and adopted in many subsequent OOD detection papers [23, 19, 8]. We did not compare to approaches that produce *classification accuracy drop* (e.g., [34, 11]), as this is a substantial limitation from a practical perspective [1]. The code to reproduce the results is available as supplementary material.

We trained many 100-layer DenseNetBCs with growth rate $k = 12$ (i.e., 0.8M parameters) [12], 110-layer ResNets [7][3], and 34-layer ResNets [7][4] on CIFAR10 [15], CIFAR100 [15], and SVHN [29] datasets with SoftMax, IsoMax, and IsoMax$_\mathcal{I}$ losses using the same procedures (e.g., initial learning rate, learning rate schedule, weight decay) presented in Lee et al. [19].

We used SGD with the Nesterov moment equal to 0.9 during 300 epochs with a batch size of 64, and an initial learning rate of 0.1 with a learning rate decay rate equal to ten applied in the epoch number 150, 200, and 250. The weight decay was 0.0001. We did not use dropout. We used a computer with CPU Intel i7-4790K, 4.00GHz, x64, octa-core, 32Gb RAM, and a GPU Nvidia GTX 1080 Ti.

---

[3]`https://github.com/akamaster/pytorch_resnet_cifar10`
[4]`https://github.com/pokaxpoka/deep_Mahalanobis_detector`

Table 4: **Unfair comparison with approaches that use input preprocessing and produce slow/inefficient inferences in addition to requiring validation using adversarial examples.** ODIN and Mahalanobis were applied to models trained using SoftMax loss. These approaches present at least four times slower and less power efficient inferences [27], as they use input preprocessing. Their hyperparameters were validated using adversarial examples. IsoMax$_\mathcal{I}$+MDS (ours) means training using IsoMax$_\mathcal{I}$ loss and performing OOD detection using minimum distance score (MDS). The best results are in bold (0.5% tolerance).

| Model | Data (training) | OOD (unseen) | Comparison with approaches that use input preprocessing and adversarial validation. | |
| --- | --- | --- | --- | --- |
| | | | AUROC (%) [↑] | DTACC (%) [↑] |
| | | | ODIN / Mahalanobis / IsoMax$_\mathcal{I}$+MDS (ours) | |
| DenseNetBC100 | CIFAR10 | SVHN | 92.8 / 97.6 / **99.5** | 86.5 / 92.6 / **96.3** |
| | | TinyImageNet | 97.2 / **98.8** / 98.6 | 92.1 / **95.0** / 93.9 |
| | | LSUN | 98.5 / **99.2** / **99.1** | 94.3 / **96.2** / 95.2 |
| | CIFAR100 | SVHN | 88.2 / 91.8 / **96.5** | 80.7 / 84.6 / **90.0** |
| | | TinyImageNet | 85.3 / 97.0 / **97.6** | 77.2 / **91.8** / **91.6** |
| | | LSUN | 85.7 / **97.9** / 97.4 | 77.3 / **93.8** / 90.8 |
| ResNet34 | CIFAR10 | SVHN | 86.5 / 95.5 / **98.2** | 77.8 / 89.1 / **93.0** |
| | | TinyImageNet | 93.9 / **99.0** / 94.8 | 86.0 / **95.4** / 88.5 |
| | | LSUN | 93.7 / **99.5** / 96.6 | 85.8 / **97.2** / 91.0 |
| | CIFAR100 | SVHN | 72.0 / 84.4 / **88.3** | 67.7 / 76.5 / **82.6** |
| | | TinyImageNet | 83.6 / 87.9 / **90.5** | 75.9 / **84.6** / **84.4** |
| | | LSUN | 81.9 / 82.3 / **88.3** | 74.6 / 79.7 / **82.6** |

Table 5: **Unfair comparison of outlier exposure-enhanced SoftMax loss with IsoMax loss and IsoMax$_\mathcal{I}$ loss without using extra data.** SoftMax$^{OE}$+ES means training using SoftMax loss enhanced during training by using outlier exposure [10], which requires the collection of outlier data, and performing OOD detection using the entropic score (ES). We used the same outlier data used in [10]. In each case, we collected the same amount of outlier data as the number of training examples present in the training set used to train SoftMax$^{OE}$. Despite being possible [26], the IsoMax loss and IsoMax$_\mathcal{I}$ loss were not enhanced with outlier exposure to keep the solution seamless. IsoMax+ES means training using IsoMax loss and performing OOD detection using the entropic score (ES). IsoMax$_\mathcal{I}$+MDS (ours) means training using IsoMax$_\mathcal{I}$ loss and performing OOD detection using minimum distance score (MDS). The values of the performance metrics TNR@TPR95 and AUROC were averaged over all out-of-distribution. The best values are in bold (0.5% tolerance).

| Model | Data (training) | Comparison of IsoMax loss variants without using extra data with outlier exposure-enhanced SoftMax loss. | |
| --- | --- | --- | --- |
| | | TNR@TPR95 (%) [↑] | AUROC (%) [↑] |
| | | SoftMax$^{OE}$+ES / IsoMax+ES / IsoMax$_\mathcal{I}$+MDS (ours) | |
| DenseNetBC100 | CIFAR10 | 93.8 / 84.1 / **95.0** | 98.5 / 97.3 / **99.1** |
| | CIFAR100 | 23.0 / 45.1 / **82.5** | 80.5 / 91.9 / **97.0** |
| ResNet110 | CIFAR10 | **92.6** / 76.5 / 81.8 | **98.0** / 96.0 / 97.0 |
| | CIFAR100 | 36.1 / 25.1 / **40.2** | 83.2 / 85.5 / **87.7** |

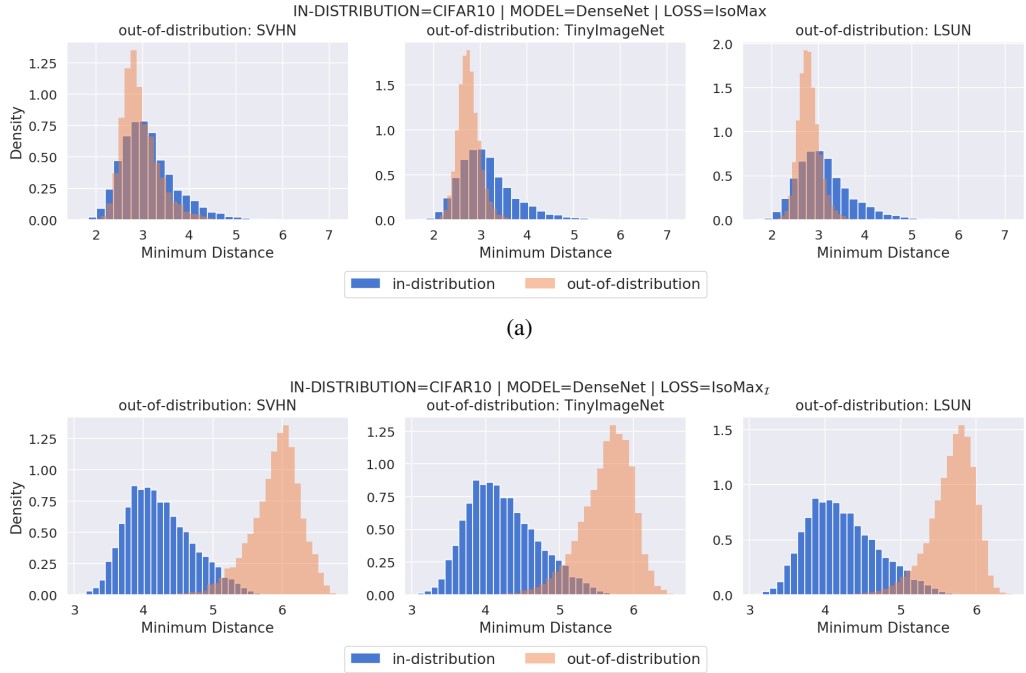

Figure 1: (a) The *no isometric distances* used by the IsoMax loss make detecting out-of-distribution examples difficult using the minimum distance score. Consequently, the minimum distance score is not competitive with the entropic score in this case. (b) The *isometric distances* used by the IsoMax$_\mathcal{I}$ loss make detecting out-of-distribution examples easy using the minimum distance score. Consequently, the minimum distance score usually overcomes the entropic score in this situation.

We used resized images from the datasets TinyImageNet [2][5] and the Large-scale Scene UNderstanding dataset (LSUN) [36][5] following Lee et al. [19] to create out-of-distribution samples. We added these out-of-distribution images to the validation sets presented in the CIFAR10, CIFAR100, and SVHN to form the test sets and evaluate the OOD detection performance.

We evaluated the OOD detection performance using the true negative rate at 95% true positive rate (TNR@TPR95), the area under the receiver operating characteristic curve (AUROC), and the detection accuracy (DTACC), which corresponds to the maximum classification probability over all possible thresholds $\delta$:

$$1 - \min_{\delta} \left\{ P_{\text{in}}\left(o\left(\mathbf{x}\right) \leq \delta\right) P\left(\mathbf{x} \text{ is from } P_{\text{in}}\right) + P_{\text{out}}\left(o\left(\mathbf{x}\right) > \delta\right) P\left(\mathbf{x} \text{ is from } P_{\text{out}}\right) \right\},$$

where $o(\mathbf{x})$ is the OOD detection score. It is assumed that both positive and negative samples have an equal probability of being in the test set, i.e., $P\left(\mathbf{x} \text{ is from } P_{\text{in}}\right) = P\left(\mathbf{x} \text{ is from } P_{\text{out}}\right)$. All the mentioned metrics follow the calculation procedures specified in Lee et al. [19].

## 4 Results and Discussion

**Classification Accuracy**   Table 2 presents the classification accuracy results. It shows that IsoMax$_\mathcal{I}$ loss does not present *classification accuracy drop* compared to SoftMax loss or IsoMax loss for all datasets and models. We observe that the IsoMax loss variants present more than one percent (%1) better accuracy than the SoftMax loss when using ResNet110 on the CIFAR100 dataset.

**Out-of-Distribution Detection**   We report the results using the entropic score for SoftMax loss (SoftMax+ES), outlier exposure-enhanced SoftMax loss (SoftMax$^{\text{OE}}$+ES), and IsoMax loss (IsoMax+ES) because it always overcame the maximum probability score and minimum distance score in

---

[5]https://github.com/facebookresearch/odin

these cases. For IsoMax$_\mathcal{I}$, we report the values using the minimum distance score (IsoMax$_\mathcal{I}$+MDS), as it usually overcame the maximum probability and the entropic score in this situation.

The Table 3 summarizes the results of the *fair* OOD detection comparison. In the mentioned table, all approaches are accurate (no *classification accuracy drop*), fast and power-efficient (inferences are performed without *input preprocessing*), and no validation is required to define hyperparameters. Additionally, no additional/extra/outlier data are needed. In most cases, IsoMax$_\mathcal{I}$+MDS overcomes IsoMax+ES performance, regardless of the model, dataset, and out-of-distribution.

The minimum distance score produces high OOD detection performance when combined with the IsoMax$_\mathcal{I}$, which evidences that the isometrization of the distances indeed work in this case. However, the same minimum distance score produced low OOD detection performance when combined with the original IsoMax loss. The Fig. 1 provides an explanation for this fact.

Table 4 summarizes the results of an *unfair* OOD detection comparison, as the methods present different requirements and produce distinct side effects. ODIN [23] and the Mahalanobis [19] approaches require adversarial samples to be generated to validate hyperparameters for each combination of dataset and model. Moreover, these approaches use *input preprocessing*, *which makes inferences at least four times slower and at least four times less energy-efficient*. Validation using adversarial examples may be a cumbersome procedure to be performed from scratch on novel datasets, as hyperparameters such as optimal adversarial perturbations may be unknown in such cases. IsoMax$_\mathcal{I}$+MDS does not present these special requirements and does not produce the mentioned side effects.

Nevertheless, IsoMax$_\mathcal{I}$+MDS provides higher performance than ODIN. Usually, this occurs by a large margin. In addition to the changes between the entropy maximization trick and temperature calibrations present in [27, 26], we emphasize that training with entropic scale affects the learning of all weights while changing the temperature during inference affects only the last layer. Hence, the fact that the proposed solution overcomes ODIN by a safe margin is additional evidence that the *entropy maximization trick often produces much higher OOD detection performance than temperature calibration, even when the latter is combined with input preprocessing. Besides, the entropy maximization trick does not require access to validation data to tune the temperature*. In addition to being seamless and avoiding the Mahalanobis approach drawbacks, IsoMax$_\mathcal{I}$+MDS usually overcomes it in terms of AUROC and produces similar performance when considering the DTACC.

Table 5 *unfairly* compares the performance of the proposed approach with the outlier exposure solution. Similar to IsoMax variants, the outlier exposure approach does not require hyperparameters tuning and produces efficient inferences. However, it requires collecting outlier data, while our approach does not. It is important to emphasize that outlier exposure may also be combined with IsoMax loss variants to increase the OOD detection performance further [26]. Nevertheless, in the mentioned table, we preferred to present the IsoMax loss variants without outlier exposure to show that the outlier exposure-enhanced SoftMax loss usually present lower OOD detection than IsoMax$_\mathcal{I}$+MDS *even without using outlier exposure*.

# 5 Conclusion

In this paper, we improved the IsoMax loss by replacing its original distance with what we call the *isometric distance*. Additionally, we proposed a zero computational cost minimum distance score. The experiments showed that these modifications produce higher OOD detection performance while keeping desired benefits of IsoMax loss (absence of hyperparameters to tune, no reliance on additional/extra/outlier data, fast and power-efficient inference, and no *classification accuracy drop*).

Similar to IsoMax loss, after training using the proposed IsoMax$_\mathcal{I}$ loss, we may apply inference-based approaches (e.g., Gram matrices, outlier exposure, energy-based) to the pretrained model to eventually increase even more the overall OOD detection performance. Therefore, *instead of competitors, the OOD detection approaches that may be applied to pretrained models are actually complementary to our approach* [27, 26]. Hence, there is no drawback in training a model using IsoMax$_\mathcal{I}$ loss instead of SoftMax loss or IsoMax loss, regardless of planning to subsequently use an inference-based OOD detection approach to increase the OOD detection performance further.

In future works, considering its simplicity, we plan to verify whether our approach scales satisfactorily to large-scale image datasets such as ImageNet. We also intend to verify the performance of our solution using text datasets.

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
