# OpenReview forum: "Improving Entropic Out-of-Distribution Detection using Isometric Distances and the Minimum Distance Score"
_NeurIPS.cc/2021/Conference — NeurIPS 2021 Submitted_

### Official Review · Reviewer_J9aE · 2021-07-13

**Rating:** 6
**Confidence:** 4

**Summary:**

The work proposes a method to improve the entropic out-of-distribution detection method which comprises the IsoMax loss for training and the entropic score for out-of-distribution detection. The IsoMax method defines the probability (also called entropic score, ES) based on the softmax of the distance from the feature map f(x) to the learnable prototype associated with each class \mu_j, i.e. p(y=k|x) = exp(-|f(x)-\mu_k|)/\sum_j exp(-|f(x)-\mu_j|). The authors observed that since the feature map f(x) are unnormalized, examples with low norms are unjustifiably favored to be considered OOD examples. So they propose to use the normalized feature map f’(x)=f(x)/|f(x)| to fix it. The method is called IsoMax_I. They also propose to use the minimum distance score min_j |f’(x)-\mu'_j| as the OOD score instead of using the maximum probability max p(y|x).

**Limitations And Societal Impact:**

Yes.

**Main Review:**

- Originality: the work proposes a simple but effective fix to the original IsoMax loss.

- Quality: the paper is written well and easy to follow. There are a few comments:
1. The experiments compare the performance among the regular softmax loss, IsoMax+ES and IsoMax_I+MDS (the proposed). It would be better to show the performance of IsoMax+MDS and IsoMax_I+ES, so that the contribution made by the two proposed components, IsoMax_I and MDS, can be disentangled. Similarly, in Figure 1, it would be great to show similar histograms for IsoMax + MDS and IsoMaxI+ES.
2. The proposed IsoMax_I loss involves a hyper-parameter called distance scale d_s. How do you tune d_s? Do you need extra validation data for tuning?

- Clarity: in general the paper's text, figures and tables are presented clearly. There are a few comments:
1. The authors mentioned that Mahalanobis method (Table 1) requires hyperparameter tuning and requires adversarial samples (line 183). In my understanding, the vanilla Mahalanobis method fits class conditional Gaussian distributions based on the feature map, which does not involve hyperparameters or adversarial samples.
2. In Table 5, it is unknown which OOD datasets are the results for.

- Significance: the results are important.


***Update: Thank the authors for your response to my questions and your efforts on providing more results. Overall the method is simple and effective. I increase my score from 5 to 6. Please include the updated figures and results into the final version. Thank you. ***

**Time Spent Reviewing:**

3

---

> ### Author Response · Authors · 2021-08-05
> **The IsoMax-I easily overcomes the vanilla Mahalanobis method in addition to avoiding feature extraction and training additional models. Therefore, in the paper, we compared against the complete Mahalanobis approach.**
>
> We sincerely appreciate the comments of the reviewer.
>
> The reviewer wrote:
>
> *The authors mentioned that the Mahalanobis method (Table 1) requires hyperparameter tuning and requires adversarial samples (line 183). In my understanding, the vanilla Mahalanobis method fits class conditional Gaussian distributions based on the feature map, which does not involve hyperparameters or adversarial samples.*
>
> The reviewer is correct to say that the **vanilla** Mahalanobis method does not require hyperparameter tuning or adversarial samples. However, in Table 1, Table 4, and throughout the paper, we consider the **complete** Mahalanobis approach, which consists of the vanilla variant added by input preprocessing and feature ensemble.
>
> We consider the complete Mahalanobis approach throughout the paper because it presents much better performance than the vanilla option. To see this, please refer to the ablation study presented in Table 1 of the seminal Mahalanobis approach paper (https://papers.nips.cc/paper/2018/file/abdeb6f575ac5c6676b747bca8d09cc2-Paper.pdf) to notice that the complete Mahalanobis approach performs much better than the vanilla alternative.
>
> In the mentioned table, we see the vanilla Mahalanobis using ResNet trained on CIFAR-10, when SVHN dataset is used as OOD produces an AUROC of 93.9%. At the same time, IsoMax-I obtains 98.2% (Table 4 of our paper), even though IsoMax avoids feature extraction and training additional models. Therefore, we conclude that IsoMax-I produces much better OOD detection performance than the vanilla Mahalanobis. This may also be noticed because IsoMax-I usually slightly outperforms the complete Mahalanobis approach, which easily outperforms the vanilla Mahalanobis. Hence, considering that IsoMax-I easily overcomes the vanilla Mahalanobis, in Table 4, we preferred to compare it to the complete Mahalanobis approach.
>
> The complete Mahalanobis approach requires input preprocessing, which requires tuning the hyperparameter epsilon. Additionally, the feature ensemble uses feature maps of many different layers. Therefore, a regression model needs to be trained to calculate their relative strengths to construct an overall distance to perform OOD detection. These procedures require adversarial examples to work as a replacement to usually unavailable out-of-distribution data. For details, please see the Mahalanobis paper (https://papers.nips.cc/paper/2018/file/abdeb6f575ac5c6676b747bca8d09cc2-Paper.pdf)
>
> We thank the reviewer for this comment. We will clarify this point in the paper.
>
> Final remarks: In the above discussion, we are considering that we are applying vanilla/complete Mahalanobis to SoftMax loss pretrained models. However, naturally, we may choose to apply them to IsoMax-I loss pretrained models to start from a much better baseline OOD detection performance. We believe that applying vanilla Mahalanobis to an IsoMax-I loss pretrained model is a relatively straightforward way to probably overcome the pure IsoMax-I loss (almost) without side effects.

---

> > ### Author Response · Authors · 2021-08-14
> > **We have update the submission to clarify that we mean the full Mahalanobis approach throughout the paper**
> >
> > Dear reviewer,
> >
> > We have added the following text to the paper to make it clear in the paper that we are considering the full Mahalanobis approach:
> >
> > _Considering that our approach easily outperforms the vanilla Mahalanobis method, throughout this paper, we use the term "Mahalanobis approach'' to refer to the "full Mahalanobis approach''._

---

> ### Author Response · Authors · 2021-08-05
> **The entropic scale is constant and the same regardless of the data and model considered. Hence, our approach does not require hyperparameter tuning or extra validation data. We have experimental evidence and theoretical insights to explain why this constant value generalizes well.**
>
> The reviewer wrote:
>
> *The proposed IsoMax_I loss involves a hyper-parameter called distance scale d_s. How do you tune d_s? Do you need extra validation data for tuning?*
>
> The distance scale is learnable, so we believe the reviewer means the entropic scale, a constant scalar regardless of the data, model, and out-of-distribution. The entropic scale is a constant value used during training that is removed before inference.
>
> This constant value is used regardless of the model, in-distribution, and out-of-distribution. The value of ten was defined in the original Entropic Out-of-Distribution Detection paper (https://arxiv.org/pdf/1908.05569v1.pdf, Figure 2). The authors trained on SVHN using the constant entropic scales equal to one, three, and ten during training and removed it for inference, which constitutes "the entropy maximization trick". They used CIFAR100 as out-of-distribution. The CIFAR100 is not used as out-of-distribution in the original Entropic Out-of-Distribution Detection paper [1], in their journal version [2], or in our work.
>
> The desired behavior was observed: the entropy increase when the entropic scale used during training increases. More important: the out-of-distribution detection performance also increases. Even more important, this behavior generalizes when other data are used as OOD. Please, see Figure 2 of the **FIRST PREPRINT FROM 2019**: https://arxiv.org/pdf/1908.05569v1.pdf. Additionally, their published version showed that the constant value of 10 generalizes to all datasets, models, and out-of-distribution data.
>
> ***Please see Figure 2 and 3 of the paper [1] and Figure 3 of the paper [2]. It was experimentally observed that the out-of-distribution detection performance presents a very well-behavior dependence on the entropic scale: it increases essentially monotonically until it reaches a saturation point around ten, regardless of the model and data considered. Please, also see a brief explanation in our paper in lines 69-79. This UNIVERSAL BEHAVIOR explains why it is always possible to use ten and obtain high-performance results! Please, notice in the mentioned figure that the classification accuracy is not affected!***
>
> Considering that the entropic scale is inside an exponential function, in the journal version of the entropic out-of-distribution detection paper, the authors argue that ten is enough to obtain almost the maximum entropy possible and that increasing this value even further does not help, as the maximum possible entropy is already achieved [2] (Section IV, A). Therefore, we observe that gains in OOD detection performance increases fast in the beginning and saturates after some point. For example, consider the $d$ is a distance. Hence, $e^{-1d}$ and $e^{-3d}$ are not as small as $e^{-10d}$. However, $e^{-10d}$ and $e^{-20d}$ are too small numbers to make any difference. This behavior is monotonic in relation to the entropic scale since we are dealing with a monotonic function: the exponential. A substantial amount of experiments is added to confirm this experimentally ([2], Figure 2). This behavior is the same regardless of the dataset and model.
>
> In machine learning and deep learning, we have many examples of **GLOBAL** values that are constant (i.e., do not need to be tuned) regardless of the data and model used. For example, in the ADAM optimizer, we all use beta1 equals 0.9, and beta2 equals 0.99. When using SGD with momentum, we all use the moment equals 0.9. These are experimentally recognized as values that work well across all circumstances. Just like those values, many experiments have demonstrated that the entropic scale equals ten works well. In addition to that, we have the theoretical insights mentioned in the paper above-mentioned.
>
> **In summary, we just used the entropic scale value defined in the original papers. We do not tune it. Consequently, we do not need extra validation data for tuning it, as the same constant value is used regardless of the data and models used for training. Besides the experimental evidence, the fact that this value is inside an exponential function gives theoretical insights to justify this fact.**
>
> [1] Macêdo, D., Ren, T. I., Zanchettin, C., Oliveira, A. L. I., and Ludermir, T. B. "Entropic out-of-distribution detection." *International Joint Conference on Neural Networks (IJCNN), 2021.* preprint: https://arxiv.org/abs/1908.05569.
>
> [2] Macêdo, D., Ren, T. I., Zanchettin, C., Oliveira, A. L. I., and Ludermir, T. B. "Entropic Out-of-Distribution Detection: Seamless Detection of Unknown Examples." *Accepted for publication in IEEE Transactions on Neural Networks and Learning Systems: Special Issue on Deep Learning for Anomaly Detection.* preprint: https://arxiv.org/abs/2006.04005

---

> ### Author Response · Authors · 2021-08-06
> **We thank the reviewer for the suggestions and observations. We will add them to the final version of the paper.**
>
> The reviewer wrote:
>
> *The experiments compare the performance among the regular softmax loss, IsoMax+ES and IsoMax_I+MDS (the proposed). It would be better to show the performance of IsoMax+MDS and IsoMax_I+ES, so that the contribution made by the two proposed components, IsoMax_I and MDS, can be disentangled. Similarly, in Figure 1, it would be great to show similar histograms for IsoMax + MDS and IsoMaxI+ES.*
>
> We agree with the reviewer. We will add the suggestion to the final paper.
>
> *In Table 5, it is unknown which OOD datasets are the results for.*
>
> We wrote the following in the legend of the mentioned table:
>
> *The values of the performance metrics TNR@TPR95 and AUROC were averaged over all out-of-distribution.*
>
> However, the reviewer is correct in saying that it is unclear which out-of-distributions we are talking about.
>
> We sincerely thank the reviewer for the observation. We will make clear that we are using the same out-of-distribution used in Table 3 (SVHN, TinyImageNet, and LSUN) in the final version of the paper.

---

> > ### Author Response · Authors · 2021-08-13
> > **As requested, we made more evident in the legend of Table 5 which OOD data are being considered**
> >
> > Dear reviewer,
> >
> > As requested, we made more evident in the legend of Table 5 that the OOD considered are the SVHN, TinyImageNet, and LSUN.
> >
> > Please, see the updated table below:
> >
> > https://i.imgur.com/bIPseMa.png
> >
> > Thanks for contributing to improving our submission.

---

> > ### Author Response · Authors · 2021-08-13
> > **As requested, we added results regarding IsoMax+MDS and IsoMax-I+ES**
> >
> > Dear reviewer,
> >
> > As requested, we added results regarding IsoMax+MDS and IsoMax-I+ES.
> >
> > Please, see the **Table 6** below:
> >
> > https://i.imgur.com/KWK7AzM.png
> >
> > As we mentioned in the paper, the original IsoMax produces extremely low performance when using a distance score!
> >
> > **This is another novelty of IsoMax-I: a simple distance score produces state-of-the-art performance, making total sense with the intuition that in-distribution should be near the prototypes (more confidence). At the same time, the OOD examples should be more distant (more uncertain)!**
> >
> > Thanks for contributing to improving our submission.

---

> ### Author Response · Authors · 2021-08-13
> **Please, notice we now have results of multiples execution of the same experiment!**
>
> Please, see the post: https://openreview.net/forum?id=f4jw35Vrk6d&noteId=FRsjpDDPKG
>
> Please, see also the comments: https://openreview.net/forum?id=f4jw35Vrk6d&noteId=Qv9bHIGJQRT

---

> ### Author Response · Authors · 2021-08-14
> **The Figure 1 was updated to comply with a request from the reviewer**
>
> Regarding the following request:
>
> _Similarly, in Figure 1, it would be great to show similar histograms for IsoMax + MDS and IsoMaxI+ES._
>
> We update the Figure 1 to show both hystograms of entropy and distances. Therefore, it is now much more clear why the distance score produces very low performance when applied to the original IsoMax.
>
> Please see the new Figure 1:
>
> https://i.imgur.com/E5XQpNm.png

---

### Official Review · Reviewer_LMfd · 2021-07-14

**Rating:** 2
**Confidence:** 5

**Summary:**

This work improves "IsoMax loss" by using normalized feature vectors (and a learned scalar multiplier) when training a CNN for classification. The paper situation the new loss function in the Out-of-Distribution literature an focuses on fast inference time without needing input preprocessing.  The results generally show the IsoMax_I loss out performs similar methods.

**Limitations And Societal Impact:**

The paper is missing the broader impact section, but there is discussion throughout on reducing energy use.

**Main Review:**

Originality:
The main proposal of this work is to improve IsoMax loss by normalizing it's prediction vectors and multiplying them by a single learned parameter. This is a very minor change over IsoMax. Various distance metrics have been used for OOD, and the proposed S_{MinDistance} is very straight-forward as it's the exact metric being optimized.

Clarity: This paper well describes the OOD problem and in what situations the new proposed loss function is suitable for use.  The title is disingenuous as the method is a change to the final layer of a neural net, while the loss function is still LogLoss (or CrossEntropy ).

Quality: The evaluation procedure is generally okay for the proposed method. There is a significant amount of text/format overlap with the prior work IsoMax Loss.

Unfortunately, comparison to state-of-the-art is lacking. Specifically, Generalized-Odin [11] often outperforms the proposed method even without input preprocessing (see table 7). Additionally, Generalized-Odin is also a similar change to the final layer of a neural net.

It's not clear in what situation IsoMax_I should be used instead of Generalized-Odin.

Post rebuttal update:

While I appreciate the authors very detailed response, my primary issues have not been addressed: novelty and comparison to SOTA. The proposed work (a change to the final layer of a CNN) is not of enough novelty to be of interest to the broader community. Some valid critiques against my recommended SOTA (generalized-odin) were given, but failed to address the primary difference in OOD score (using a "fair" comparison). Disappointingly, the results discussed in the comments are cherry-picked by architecture. I've run G-odin myself and get better results than reported in this paper (without input preprocessing/hyper-param tuning). Lastly, there's significantly more datasets used in OOD papers in the last year or 2, and it's not clear if cifar10/cifar100 results generalize to harder datasets.

For these reasons I will keep my initial score.



**Time Spent Reviewing:**

4

---

> ### Author Response · Authors · 2021-08-04
> **A vehement refutation of an unfortunate false claim performed by the Reviewer LMfd**
>
> Before any additional comments, we would like to vehemently refute the bellow unfortunate false claim performed by the Reviewer LMfd:
>
> *"Specifically, Generalized-Odin [11] often outperforms the proposed method even without input preprocessing (see table 7)."*
>
> **Actually, exactly the opposite is true! Our approach IsoMax-I always outperforms the best Generalized-Odin (G-ODIN) variant in each case!**
>
> All values below for G-ODIN were collected from Table 7 mentioned by the reviewer (https://arxiv.org/pdf/2002.11297.pdf). Naturally, the IsoMax-I results are from our paper. OOD means "out-of-distribution".
>
>
> ___
> **In-Distribution CIFAR10:**
>
> OOD SVHN: G-ODIN [DeConf-I-g(x)]=99.3%, ISOMAX-I=**99.5%**
>
> OOD TinyImageNet (resized): G-ODIN [DeConf-E-h(x)]=96.9%, ISOMAX-I=**98.6%**
>
> OOD LSUN (resized): G-ODIN [DeConf-I-h(x)]=98.2%, ISOMAX-I=**99.1%**
>
>
>
> **In-Distribution CIFAR100:**
>
> OOD SVHN: G-ODIN [DeConf-C-h(x)]=93.6%, ISOMAX-I=**96.5%**
>
> OOD TinyImageNet (resized): G-ODIN [DeConf-E-h(x)]=95.5%, ISOMAX-I=**97.6%**
>
> OOD LSUN (resized): G-ODIN [DeConf-E-h(x)]=95.5%, ISOMAX-I=**97.4%**
>
>
>
> **In-Distribution SVHN:**
>
> OOD CIFAR10: G-ODIN [DeConf-I-h(x)]=98.4%, ISOMAX-I=**99.1%**
>
> OOD TinyImageNet (resized): G-ODIN [DeConf-I-h(x)]=98.7%, ISOMAX-I=**99.7%**
>
> OOD LSUN (resized): G-ODIN [DeConf-I-h(x)]=98.4%, ISOMAX-I=**99.7%**
> ___
>
>
> We always considered the performance of the best variant of G-ODIN, regardless of the in-distribution and OOD. Please, notice that the G-ODIN flavor that presents the best performance variates depend on the in-distribution and OOD considered!
>
> **In the real world, we will have to choose a particular flavor of G-ODIN for use. Therefore, the performance difference in favor of IsoMax-I will be even greater than shown above.**

---

> > ### Comment · Reviewer_LMfd · 2021-08-04
> > **g-odin discussion**
> >
> > My apologies, I should I been more specific that I was comparing the results between the Resnet architecture; Table 7 is using a ResNet 34 architecture same as your table 4 bottom half. Comparing their primary classifier head (DeConf-C-h(x) ) to your reported results shows the difference in performance I mentioned (I.E. on cifar10 vs. ood-SVHN your method's AUC is 98.2 and godin's is 98.6).
> >
> > I don't believe g-odin reports dense-net results without input preprocessing.

---

> > > ### Author Response · Authors · 2021-08-04
> > > **The problem with the many variants of the G-ODIN: Which one to use? How to decide without access to out-of-distribution?**
> > >
> > > **A major problem with the G-ODIN is that they are actually many proposals rather than just one.**
> > >
> > > For example, compared with the reviewer hand-picked option, G-ODIN is competitive with IsoMax-I. However, IsoMax-I easily overcomes other variants of the G-ODIN. How do you know beforehand which one of the G-ODIN variants performs better for a particular dataset and model? Do we need to train the same network on the same data many times with different G-ODIN variants?
> > >
> > > **The reviewer hand-picked this specific DeConf-C-h(x) G-ODIN variant (among nine options) to compete with IsoMax-I in this particular case because he saw the performance on TEST DATA! All this to present a 0.4% better performance than IsoMax-I in a particular combination of in-distribution, out-of-distribution, and model.**
> > >
> > > Please, see Table 7; for each combination of dataset and model, we have a different "best" G-ODIN flavor to select at our convenience. We have nine options to choose from! Nine times we trained the same model on the same data, just changing the G-ODIN variant! In other combinations of datasets and models in Table 7, different flavors of G-ODIN perform better than DeConf-C-h(x). **For example, on SVHN, the same DeConf-C-h(x) performs much worse than many others G-ODIN flavors. See also Figures 3a,3b,4a, and 4b. The DeConf-C-h(x), which was called by the reviewer the "primary classifier head", never presented the best results in these cases.**
> > >
> > > Moreover, for comparing with Mahalanobis in Tables 1 and 2, the authors say the following in the legend of Table 1: _"the DeConf-C is selected since it shows the best robustness in our analysis"_. It is not clear if they are talking about the plain, $g(x)$, or $h(x)$ variant of the DeConf-C. It does not matter. What matters is: _How do they know that this particular variant presents the best robustness (stable results) in this particular case?_ Answer: They saw test data, unconsciously performed model selection using it (i.e., they used test set as validation set), and finally reported validation accuracy as test accuracy (i.e., they overestimated the test performance) to compare against other approaches! Remember that test data contains out-of-distribution data, which the authors claimed was never used other than to evaluate test performance!
> > >
> > > **Even more important, the same hand-picked DeConf-C-h(x) G-ODIN variant produces LOWER results than our IsoMax-I on SVHN (unfortunately, we did not present these results in the submitted version of the paper, but we may add those to the final version). This makes us believe that IsoMax-I presents much more consistent/stable results across different datasets.**
> > >
> > > ***Please, also notice that trying to fix the DeConf-C-h(x) classification accuracy problem creates an OOD detection performance problem: https://openreview.net/forum?id=f4jw35Vrk6d&noteId=ZdJ7f1o41mO. What is the point of mitigating the drop in classification accuracy if OOD detection performance drops dramatically?***
> > >
> > > After train many neural networks using different G-ODIN flavors, how could we select the one to use? Do we need to lose more training data to validate/choose the best G-ODIN? Please, notice that the procedure to validate hyperparameters using only in-distribution data _does not indicate which variant will present the best OOD detection performance on test data!_ Therefore, we still need a way to select/validate which one to use. Can this be done without access to out-of-distribution data? _In other words, the G-ODIN paper claims that we do not need access to out-of-distribution data to use the mentioned approach. However, considering that we do not have a clear winner in all cases, how can we select/validate the G-ODIN variant to use without out-of-distribution data?_
> > >
> > > If training many networks and competing using many flavors are allowed, we may also propose training many networks with different values of entropic scale. For example, we could train also using entropic scales equals 15, 20, 25, 30, 35, 40, 45, and 50 (nine different options). After that, we look at test data and choose the best one for a particular case to compete against G-ODIN.
> > >
> > > This is the problem of providing many variants without a clear winner: we have the illusion that the solution performs well because we always intuitively select the performance of the best variant for each case by looking at the test data. However, in the real world, we only deploy a single model. Hence, we need a single approach/variant that works well regardless of the dataset, model, and out-of-distribution.
> > >
> > > **In summary, considering that there is no clear winner across data, model, and out-of-distribution, how can we decide which variant to use/deploy without access to out-of-distribution? How to validate the best G-ODIN flavor for a given situation without validating using out-of-distribution data? If, in the field, the system is subject to other out-of-distribution, are the reported estimation reliable or overestimated?**
> > >
> > > ***Therefore, we can not look at a table at the G-ODIN paper, find the best result for a particular case among the many available flavors and believe this is a reliable estimation of the OOD detection performance of the mentioned approach in the real world! Otherwise, we are doing model selection using TEST SET AND OUT-OF-DISTRIBUTION DATA!***

---

> > > ### Author Response · Authors · 2021-08-06
> > > **The fully connected layer added by G-ODIN clearly produces OVERFITTING!**
> > >
> > > ### **We made it clear in the paper that we did not compare to approaches that present classification accuracy drop because this is extremely harmful from a practical point of view [1].**
> > >
> > > ###########################
> > >
> > > ***We have a critical use case of fundamental importance in Deep Learning:***
> > > ***Training with few labeled data per class!***
> > > ***G-ODIN has a clear problem in this case.***
> > > ***Why? Because G-ODIN adds an extra fully connected layer to the end of the model, modifying the original architecture of the neural network and making it much more prone to produce overfitting!***
> > > ***Moreover, trying to solve this problem by retraining the entire model with added regularization makes the OOD detection collapses (Figures 4a and 4b)!***
> > >
> > > ###########################
> > >
> > > ***For example, considering that even without removing data from training set for validation, G-ODIN present significative classification accuracy drop of about 2% or 3% using ResNet34 on CIFAR100 that presents 500 training examples per class (last table of supplementary material), what should we expect of the classification accuracy drop using only 100 labeled training images per class? The paper [2] (Figure 3) showed that IsoMax does not produce classification accuracy drop using ResNet34 on CIFAR100 even under this severe few labeled data per class regime (CIFAR100 with just 100 labeled training images per class).***
> > >
> > > ###########################
> > >
> > > G-ODIN presents classification accuracy drop of about 3% (DeConf-C mentioned by the reviewer as G-ODIN main head) in CIFAR100. Why? We have a clear answer. G-ODIN always adds a fully connected layer to the end of the neural network! We all know that adding fully connected layers to the end of the models produces overfitting and, consequently, classification accuracy drop (i.e., see, for example, the evolution from VGG [many fully conected layers in the end of the neural network] to ResNet [many fully connected layers removed] architecture).
> > >
> > > *Please, notice that trying to mitigate this by adding extra regularization a posteriori and retraining the network all over again (i.e., the author's proposed solution) does not make much sense, as we see in Figures 4(a) and 4(b) that the OOD detection performance collapses very strongly!*
> > >
> > > #####################################################################################
> > > ___
> > > **Please, notice that CIFAR100 has 500 training examples per class. Considering that this is already a considerable problem for G-ODIN in this case, all we know about neural networks tells us that using just 100 training examples per class could only make the problem MUCH WORSE. The paper [2] (Figure 3) showed that IsoMax does not produce overfitting, even using only 100 training examples per class in CIFAR100.**
> > >
> > > ***Hence, unlike G-ODIN, IsoMax-I is a ROBUST solution, as it NEVER produces overfitting (classification accuracy drop) while ALWAYS produces state-of-the-art performance. Why? Simple answer: IsoMax-I does not add a fully connected layer to the end of the neural network, which would make it more prone to overfitting.***
> > >
> > > #####################################################################################
> > > ___
> > >
> > > So, in the paper, we preferred to compare with other non-seamless state-of-the-art approaches much more largely used such Mahalanobis and Outlier Exposure that do not present the above-mentioned methodological problems and do not present classification accuracy drop.
> > >
> > > [1] Carlini, N., Athalye, A., Papernot, N., Brendel, W., Rauber, J., Tsipras, D., Goodfellow, I. J., Madry, A., and Kurakin, A. On evaluating adversarial robustness. CoRR, abs/1902.06705, 2019.
> > >
> > > [2] Macêdo, D., Ren, T. I., Zanchettin, C., Oliveira, A. L. I., and Ludermir, T. B. "Entropic Out-of-Distribution Detection: Seamless Detection of Unknown Examples." Accepted for publication in IEEE Transactions on Neural Networks and Learning Systems: Special Issue on Deep Learning for Anomaly Detection. preprint: https://arxiv.org/abs/2006.04005.

---

> > > ### Author Response · Authors · 2021-08-06
> > > **More about the methodological problems present in the Generalized-Odin paper**
> > >
> > > The G-ODIN paper validated their hyperparameters using the CIFAR10/CIFAR100 validation sets, which were also used to evaluate the OOD detection performance! Considering we should not use the same data for both validation and test, we conclude that the OOD detection performance presented in the G-ODIN paper is probably overestimated.
> > >
> > > Additionally, the in-distribution data used to validate G-ODIN hyperparameters should have been removed from the CIFAR10/CIFAR100 training data for a fair comparison with the classification accuracy presented by SoftMax (or IsoMax-I) since SoftMax (or IsoMax-I) does not require removing training data to tune hyperparameters! Otherwise, we should be allowed to add the CIFAR10/CIFAR100 validation set to the training data for use with SoftMax/IsoMax-I.
> > >
> > > In other words, for a fair comparison of classification accuracy and OOD detection performance, we should allow training IsoMax-I adding the CIFAR10/CIFAR100 validation sets to the training data, as the data G-ODIN used for validation should be allowed to be incorporated as training data when comparing with SoftMax or IsoMax-I.
> > >
> > > Not removing validation data from the training set may be an even worse methodological problem with the G-ODIN paper than using CIFAR10/CIFAR100 validation sets for both hyperparameter tuning and OOD detection test performance evaluation. Both issues cumulatively contribute to overestimating the classification accuracy and OOD detection performance of G-ODIN presented in the paper.
> > >
> > > Hence, the validation data should have been removed from the training data, as the CIFAR10/CIFAR100 validation sets were used as test data to evaluate the classification accuracy and the OOD detection performance. By doing so, the classification accuracy drop presented by the G-ODIN approach would be even more significant (we remember that, unlike G-ODIN, IsoMax-I does neither produce classification accuracy drop nor require removing validation data from the training set.). By itself, this increased classification accuracy drop would likely contribute to another source of G-ODIN OOD detection performance drop in addition to the previously mentioned.
> > >
> > > See also the post https://openreview.net/forum?id=f4jw35Vrk6d&noteId=EzOoGTMrQTg.
> > >
> > > Please, see lines 49-59 of our paper for a brief explanation of the above discussion.

---

> > > ### Author Response · Authors · 2021-08-09
> > > **When you try to fix the classification accuracy drop of almost 3% of DeConf-C-h(x) (i.e., the G-ODIN variant that the reviwer called "the primary classifier head") by adding regularization as suggested by the authors, the OOD detection performance plummets very substantially!**
> > >
> > > The reviewer says that the DeConf-C-h(x) is the "primary classifier head" of the G-ODIN solution.
> > >
> > > ***However, the last table of the supplementary material of the G-ODIN paper CLEARLY shows that a classification accuracy drop of almost 3% using ResNet34 (exactly the example mentioned by the reviewer) is observed on CIFAR100. The reviewer never mentions this fact! He complete ignores this advantage of IsoMax-I (i.e., IsoMax-I never present classification accuracy drop) in relation to baseline.***
> > >
> > > The authors propose to fix the problem by adding regularization with a very unusual dropout with p=0.7. After that, the classification accuracy drop is reduced (not entirely solved) to about 1%.
> > >
> > > ################################################
> > >
> > > ***Here comes the really worst part: when you try to fix the classification accuracy problem (drop of about 3% drop) when using ResNet34 on CIFAR100, the OOD detection performance of DeConf-C-h(x) mentioned by the reviewer plunges very significantly (about 10%)!***
> > >
> > > ################################################
> > >
> > > Please, read what the authors wrote: _"Lastly, we note that the DeConf-E and DeConf-C have a reduced performance with extra regularization in Figure 4b."_
> > >
> > > ***Please, see the Figures 4(a) and 4(b). Trying to fix the classification accuracy drop by adding extra regularization (dropout p=0.7) made the OOD detection performance of the DeConf-C-h(x) nosedives about 10% (from 94.2% to 84.8%)! Hence, when you try to mitigate (without complete success) the classification accuracy drop problem of DeConf-C-h(x) using ResNet34 on CIFAR100, the G-ODIN approach is no longer competitive in terms of OOD detection performance.***
> > >
> > > ##################################################
> > >
> > > ***Therefore, the DeConf-C-h(x) using ResNet34 (exactly the example hand-picked by the reviewer) can not avoid a severe classification accuracy drop of about 3% on CIFAR100 and simultaneously present competitive OOD detection performance.***
> > >
> > > ##################################################

---

> ### Author Response · Authors · 2021-08-04
> **In our opinion, IsoMax-I should always be used instead of Generalized-Odin (PART I)**
>
> The review wrote:
>
> *"It's not clear in what situation IsoMax_I should be used instead of Generalized-Odin."*
>
> Considering the reasons below, we see no reason to use Generalized-Odin (G-ODIN) instead IsoMax-I.
> ___
> **THE G-ODIN CLASSIFICATION ACCURACY DROP PROBLEM IS EVEN WORSE THAN REPORTED IN THE PAPER (TABLE 8 IN ARXIV VERSION {TABLE 4 OF SUPPLEMENTARY MATERIAL IN THE OFFICIAL CVPR VERSION} [IN BOTH CASES THE LAST TABLE OF THE SUPPLEMENTARY MATERIAL] OF G-ODIN PAPER, CIFAR 100 RESULTS):**
>
> https://arxiv.org/pdf/2002.11297.pdf
>
> https://openaccess.thecvf.com/content_CVPR_2020/supplemental/Hsu_Generalized_ODIN_Detecting_CVPR_2020_supplemental.pdf
>
> ***First, please notice in the mentioned tables that all variants presented classification accuracy drop of almost 3% on CIFAR100 dataset using ResNet34 (in other models, the difference is about 2%).***
>
> **Additionally, unlike SoftMax and IsoMax-I, G-ODIN requires reserving some in-distribution data to use as the in-distribution validation set to tune their hyperparameters, which necessarily reduces the in-distribution data available for training when compared with SoftMax and IsoMax-I. Hence, considering that the G-ODIN paper used the CIFAR10/CIFAR100 validation sets for construction of the OOD detection TEST SET, they should have removed the required in-distribution validation data from the CIFAR10/CIFAR100 training set.**
>
> Unfortunately, rather than reducing their CIFAR10/CIFAR100 training sets (which would probably increase even more the classification accuracy drop presented in some cases in G-ODIN paper), they used the CIFAR10/CIFAR100 validation set to validate the required hyperparameters. Indeed, considering that G-ODIN required 10.000 examples for validation (the size of the CIFAR10/CIFAR100 validation set), for a fair classification accuracy comparison against SoftMax and IsoMax, they should have removed this amount of data from the 50.000 training set, otherwise both SoftMax and IsoMax should be allowed to training using the additional 10.000 examples presented in the CIFAR10/CIFAR100 validation sets.
>
> Even without following the above procedure for a fair comparison with SoftMax loss classification accuracy, which is similar to the IsoMax loss classification accuracy, the authors presented some cases of classification accuracy drop in the paper (CIFAR100 results in the above-mentioned tables). To try to circumvent this severe limitation, the authors sometimes added dropout to try to avoid the classification accuracy drop.
>
> **After retraining the network using dropout with this very fine-tuned 0.7 value, they reduced the classification accuracy drop from 3% to about 1% on CIFAR100 using ResNet34. However, the OOD detection performance plummeted very strongly (https://openreview.net/forum?id=f4jw35Vrk6d&noteId=ZdJ7f1o41mO), which make it easily outperformed by IsoMax-I. We would appreciate some comments from the reviewer about this.**
>
> Why dropout specifically with p=0.7? Did the authors try many other values? Will this value also reduce the classification accuracy drop when dealing with other datasets and models? Even worse, does it make sense to mitigate the classification accuracy drop while making the OOD detection performance plummet (Figures 4a and 4b)? Do you have to always also train the baseline to realize if the classification accuracy drop happened? The IsoMax-I loss does not produce classification accuracy drop. Actually, we observed the opposite: IsoMax-I sometimes improved the classification accuracy in relation to SoftMax.
>
> ***Finally, please notice that trying to fix the classification accuracy problem as suggested by the authors themselves creates a severe OOD detection performance problem (https://openreview.net/forum?id=f4jw35Vrk6d&noteId=ZdJ7f1o41mO). Hence, does this classification accuracy drop mitigation make sense?***
>
> ___
> **G-ODIN OUT-OF-DISTRIBUTION DETECTION PERFORMANCE IS PROBABLY OVERESTIMATED IN THE PAPER:**
>
> Considering the authors used the CIFAR10/CIFAR100 validation set for **BOTH** hyperparameter tuning (i.e., used as validation data) and out-of-distribution detection performance evaluation (i.e., also used as test data), it is reasonable to argue that the out-of-distribution detection performances presented in the G-ODIN paper may be overestimated.
>
> **We all know that we are not allowed to use the same data to simultaneously validate the hyperparameters of our solution and to test our performance.**
>
> Besides that, correcting the methodological problem mentioned in the previous item would probably decrease the classification accuracy and consequently also negatively impact in the OOD detection performance. Please, notice that we are talking about two different and independent cumulative sources of OOD detection performance overestimation.
> ___
>
> ***Please, notice that this discussion is briefly presented in lines 49-59 of our paper.***
>
> See also the post https://openreview.net/forum?id=f4jw35Vrk6d&noteId=Dg724Ga55E5.

---

> > ### Author Response · Authors · 2021-08-04
> > **In our opinion, IsoMax-I should always be used instead of Generalized-Odin (PART II)**
> >
> > The review wrote:
> >
> > *"It's not clear in what situation IsoMax_I should be used instead of Generalized-Odin."*
> >
> > Considering the reasons below, we see no reason to use Generalized-Odin (G-ODIN) instead IsoMax-I.
> > ___
> > **THE ISOMAX-I, WHICH IS CURRENTLY THE ONLY APPROACH THAT WORKS AS A SEAMLESS SOFTMAX DROP-IN REPLACEMENT, PRODUCES COMPETITIVE STATE-OF-THE-ART OUT-OF-DISTRIBUTION DETECTION PERFORMANCE:**
> >
> > ########################################################################
> >
> > ########################################################################
> >
> > **In-Distribution CIFAR10:**
> >
> > **OOD SVHN:** G-ODIN=98.8%, ISOMAX-I=99.5% ***(DIFFERENCE: 0.7% ISOMAX-I)***
> >
> > **OOD TinyImageNet (resized):** G-ODIN=99.1%, ISOMAX-I=98.6 (DIFFERENCE: 0.5% G-ODIN)
> >
> > **OOD LSUN (resized):** G-ODIN=99.4%, ISOMAX-I=99.1% (DIFFERENCE: 0.3% G-ODIN)
> >
> > _MEAN DIFFERENCE: 0.03% IN FAVOR OF G-ODIN USING HYPERPARAMETER TUNING AND INEFFICIENT INFERENCES (INPUT PREPROCESSING). ADDING INPUT PREPROCESSING TO ISOMAX-I WOULD PROBABLY MAKE ISOMAX-I OUTPERFORM G-ODIN._
> >
> > **In-Distribution CIFAR100:**
> >
> > **OOD SVHN:** G-ODIN=95.9%, ISOMAX-I=96.5% ***(DIFFERENCE: 0.6% ISOMAX-I)***
> >
> > **OOD TinyImageNet (resized):** G-ODIN=98.6%, ISOMAX-I=97.6% (DIFFERENCE: 1.0% G-ODIN)
> >
> > **OOD LSUN (resized):** G-ODIN=98.7%, ISOMAX-I=97.4% (DIFFERENCE: 1.3% G-ODIN)
> >
> > _MEAN DIFFERENCE: ABOUT 0.5% IN FAVOR OF G-ODIN USING INEFFICIENT INFERENCES (INPUT PREPROCESSING) AND HYPERPARAMETER TUNING. ADDING INPUT PREPROCESSING TO ISOMAX-I WOULD PROBABLY MAKE ISOMAX-I OUTPERFORM G-ODIN._
> >
> > ########################################################################
> >
> > ########################################################################
> >
> > ***Please, see the table above (results from Table 4 of our paper and Table 2 of the G-ODIN paper). In the mentioned table, G-ODIN is using input preprocessing (i.e., producing 4X slower and also 4X less energy efficient inferences [https://arxiv.org/pdf/1908.05569.pdf, Table IV]), hyperparameter tuning, and producing classification accuracy drop on CIFAR100. Sometimes IsoMax-I outperforms G-ODIN, sometimes G-ODIN outperforms IsoMax-I. Even using input preprocessing and requiring hyperparameter tuning, in the cases in which G-ODIN outperforms IsoMax-I, it usually does by 1% or less! Adding the input preprocessing to IsoMax-I would probably reduce this difference or could even make IsoMax-I overcomes the G-ODIN also in these cases.***
> >
> > ***Moreover, considering the clear sources of overestimation presented in the G-ODIN paper explained in other posts (https://openreview.net/forum?id=f4jw35Vrk6d&noteId=Dg724Ga55E5, https://openreview.net/forum?id=f4jw35Vrk6d&noteId=EzOoGTMrQTg), we could say that whether we correctly calculated the G-ODIN performance by removing all sources of overestimation, the IsoMax-I would outperform G-ODIN in all cases, perhaps even without using input preprocessing! One last observation: the above comparison is against the G-ODIN variant that the authors hand-picked after seeing the performance of all variants on TEST DATA. In other words, they did not realize it, but they actually used, unconsciously, out-of-distribution data to do this, which naturally also produces overestimation! Please, see also: https://openreview.net/forum?id=f4jw35Vrk6d&noteId=-GY5DvsmVYX***
> >
> > If the reviewers desire, we may add the above table and comments to the final version of the paper. However, we prefer not to because publishing or reproducing overestimated results does not contribute too much to advance our research field.
> >
> > Regardless of this discussion regarding the final performance, we have always to keep in mind that IsoMax-I presents no classification accuracy drop, no input preprocessing, and no hyperparameter tuning. We believe that adding input preprocessing is not sustainable from an environmental and cost point of view, as it increases energy consumption and inference delay.
> >
> > ***Despite competitive OOD detection performance, G-ODIN presents classification accuracy drop of almost 3% (the DeConf-C variant mentioned by the reviewer) on CIFAR100 using ResNet34 (last table of supplementary material). Even worse, when you try to fix this using the solution proposed by the authors themselves, the OOD detection performance plummets very significantly. Consequently, IsoMax-I outperforms G-ODIN in such case (https://openreview.net/forum?id=f4jw35Vrk6d&noteId=ZdJ7f1o41mO).***
> >
> > ############################################################
> >
> > ***Hence, even considering the best head mentioned by the reviewer, G-ODIN is not simultaneously competitive with IsoMax-I in both classification accuracy and OOD detection performance using ResNet34 on CIFAR100! If G-ODIN is competitive in one task, it is outperformed in the other!***
> >
> > ############################################################
> >
> > ___
> > **ISOMAX-I IS MUCH EASIER TO USE THAN G-ODIN, AS THE FORMER (A SOFTMAX LOSS DROP-IN REPLACEMENT) DOES NOT REQUIRE REDUCING TRAINING DATA (WITH NEGATIVE IMPACT IN CLASSIFICATION ACCURACY) TO CREATE A VALIDATION SET TO TUNE HYPERPARAMETERS:**
> >
> > By avoiding using in-distribution validation data to tune hyperparameters, IsoMax is much easier to use than G-ODIN. In PyTorch pseudocode, we need to change just two lines of code:
> >
> > First, we need to replace the model classifier last layer with the Isometric IsoMax loss first part:
> >
> > ```python
> > class Model(nn.Module):
> >     def __init__(self):
> >     (...)
> >     #self.classifier = nn.Linear(num_features, num_classes)
> >     self.classifier = losses.IsoMaxIsometricLossFirstPart(num_features, num_classes)
> > ```
> >
> > Second, we need to replace the criterion by the Isometric IsoMax loss second part:
> >
> > ```python
> > model = Model()
> > (...)
> > #criterion = nn.CrossEntropyLoss()
> > criterion = losses.IsoMaxIsometricLossSecondPart(model.classifier)
> > ```
> >
> > ___
> > ########################################################
> >
> > **UNLIKE ISOMAX-I, G-ODIN REQUIRES CHANGES IN TRAINING/OPTIMIZATION:**
> >
> > Indeed, unlike IsoMax-I, the person that plans to use G-ODIN needs to pay attention to modifying the training/optimization of the network because the weight decay needs to be disabled to some components of the G-ODIN solution.
> >
> > ########################################################
> >
> >  ___
> > ########################################################
> >
> > **UNLIKE ISOMAX-I, G-ODIN ADDS AN EXTRA COMPOUND LAYER TO THE ARCHITECTURE:**
> >
> > Indeed, unlike IsoMax-I, G-ODIN requires adding an extra compound layer $g(x)$ to the solution with a consequent increased total number of parameters. This compound layer comprises a sigmoid function, batch normalization, and a linear layer. We do not know how well this architecture modification will generalize well to other data and models not used in the paper regarding classification accuracy and OOD detection performance.
> >
> > ########################################################
> >
> > ___
> > ***Final remarks: Considering the reviewers did not find anything technically wrong with our approach (actually, the opposite is true, as majority saw a robust proposal with competitive state-of-the-art performance), we suggest accepting the paper and let people decide by themselves which one (G-ODIN, IsoMax-I, or a third option) they prefer to use in practice in the real world.***

---

> > > ### Author Response · Authors · 2021-08-10
> > > **In our opinion, IsoMax-I should always be used instead of Generalized-Odin [G-ODIN] (FINAL REMARKS)**
> > >
> > > ***####################################################################################################***
> > >
> > > ***####################################################################################################***
> > >
> > > ***In summary, we believe we should always prefer to use IsoMax-I for the following reasons:***
> > >
> > > ***1. The OOD detection performance of both IsoMax-I and the best G-ODIN flavor chosen by the reviewer are extremely similar. Sometimes IsoMax-I outperforms G-ODIN. Sometimes we observe the opposite. However, IsoMax-I NEVER presents classification accuracy while G-ODIN sometimes presents SEVERE classification accuracy drop. Even worse, trying to mitigate this makes the OOD detection performance plummet.***
> > >
> > > ***2. IsoMax-I is extremely easier to understand, implement, and use than G-ODIN.***
> > >
> > > ***3. When using G-ODIN, we may need to retrain the network all over again to mitigate an eventual classification accuracy drop. Additionally, we must always also train the baseline approach to compare the classification accuracy and verify whether the classification accuracy drop problem is present, as G-ODIN sometimes produces overfitting. Hence, rather than training a model once, as it happens when using IsoMax-I, we need to train the same model twice or even three times when using Generalized-Odion (G-ODIN). Now comes the worst part: Training the same model many times does not actually solve the problem, as the OOD detection performance collapses (Please, see topics 4 and 8 below).***
> > > ___
> > >
> > > ***4. It is reasonable to expect that the classification accuracy drop will be greater than the 3% observed training ResNet34 on CIFAR100 when using G-ODIN to deal with datasets with fewer than 500 examples per class (see also: https://openreview.net/forum?id=f4jw35Vrk6d&noteId=Y0ByITe-Sqx), as the additional layers will contribute even more to produce overfitting. Why does G-ODIN produces overfitting? Because it adds an additional fully connected layer to the end of the neural network! Hence, it does NOT present robust behavior in the very important few labeled data per class regime.***
> > > ___
> > >
> > > ***5. When using the G-ODIN, we do not know if adding dropout with precisely p=0.7 will mitigate the problem in other situations. Please, notice that the classification accuracy drop was not fixed, as we observe a persistent classification accuracy drop of about 1% even after adding extra regularization. Hence, G-ODIN is NOT ROBUST, as it presents cases in which it can NOT provide competitive classification accuracy and simultaneously high OOD detection performance.***
> > >
> > > ***6. Moreover, the proposal to mitigate the classification accuracy drop of G-ODIN does not make much sense, as the OOD detection performance plummeted substantially in such cases. In other words, unlike IsoMax-I, G-ODIN can not simultaneously avoid classification accuracy drop and present high OOD detection performance in all situations.***
> > >
> > > ***7. When using G-ODIN, we need to change the training/optimization to avoid weight decay to some components of the G-ODIN solution.***
> > > ___
> > > ######################
> > > ___
> > > ***8. It is possible that modifying the neural network architecture by adding the compound layer required by G-ODIN does not generalize well to other models and datasets. For example, it may increase overfitting. By the way, this may very well explain the G-ODIN classification accuracy drop problem, as G-ODIN adds too many new learnable parameters in a critical location: the last layer! Indeed, we all know that the model becomes much more flexible and tends to overfit when we add extra fully connected layers to its end. Trying to minimize the overfitting a posteriori by adding regularization makes the OOD detection performance collapse (https://openreview.net/forum?id=f4jw35Vrk6d&noteId=ZdJ7f1o41mO). IsoMax-I only adds a single new learnable parameter: the distance scale. We need always to remember: Simple solutions have a much better chance to generalize well.***
> > > ___
> > > ######################
> > > ___
> > > ***9. IsoMax-I does not require hyperparameter tuning.***
> > >
> > > ***10. IsoMax-I does not produce inefficient inferences.***
> > >
> > > ***11. We do not have many flavors of the IsoMax-I to choose from.***
> > >
> > > ***12. When using IsoMax-I, the model is immediately available after training to perform OOD detection, while G-ODIN requires an extra post-training phase to tuning hyperparameters.***
> > >
> > > ***13. It is hard to know in advance which G-ODIN variant will perform better for a particular model on a given dataset.***
> > >
> > > ***We would appreciate it whether the reviewer considers the above comments and reevaluates the score.***
> > >
> > > ***####################################################################################################***
> > >
> > > ***####################################################################################################***
> > >
> > > Albert Einstein: _"Everything Should Be Made as Simple as Possible, But Not Simpler."_

---

> ### Author Response · Authors · 2021-08-06
> **Regarding calling the Isometric IsoMax (IsoMax-I) a loss rather than a output layer**
>
> We call the Isometric IsoMax a loss rather than an output layer because the entropic scale is present during the network training but removed before inference (i.e., the Entropy Maximization Trick). **Hence, the entropic scale clearly does not belong to the output layer, as it is not present during inference.**
>
> Additionally, we mentioned in the paper that we followed the terminology defined in the paper [1] (and followed by many subsequent papers) that collectively calls the combination of the last layer, the SoftMax activation function, and the cross-entropy loss as "the SoftMax loss."
>
> **Please, see the Figure 1 and Equation 1 of [1].**
>
> Please, also see lines 45-46 of our paper and the reference [1]. Hence, we made the used terminology clear in the paper.
>
> Think of a generalized loss as something applied directly to the final high-level feature layer of the neural network of the encoder model, as proposed in [1] and adopted subsequently by many papers. This mindset is one of the core aspects that led to developing the IsoMax variants to tackle the OOD detection problem, as the designer has more flexibility to construct a powerful solution.
>
> For example, in the self-supervision research area, please notice that they also use this expanded concept of loss. See, for example, the equation (1) in: https://arxiv.org/pdf/2004.11362.pdf. The mentioned equation is simply called a baseline "Self-Supervised Contrastive Loss." **This combined concept is a modern and much more powerful way to refer to (and to design) a loss.** Please, see also the equations 2 and 3. The log (strictly speaking, the cross entropy loss) is just a tiny piece of the "Supervised Contrastive Losses".
>
> Finally, as we also plan to modify the cross-entropy term itself (the log function) in future works, we have more one reason to prefer to follow the terminology in [1] by collectively call the combination of the last layer, the SoftMax activation function, and the cross-entropy simply as a loss.
>
> [1] Liu, W., Wen, Y., Yu, Z., and Yang, M. Large-margin **Softmax Loss**for Convolutional Neural Networks. International Conference on Machine Learning, 2016.

---

> > ### Author Response · Authors · 2021-08-15
> > **More important than using modern terminology is the fact that the proposed solution presents many advantages in relation to G-ODIN**
> >
> > Regardless of the modern terminology for the concept of a loss used in our paper, what really matters is that the proposed solution present many advantages in relation to G-ODIN.
> >
> > Please see https://openreview.net/forum?id=f4jw35Vrk6d&noteId=GvT-H_VMIEK.

---

> ### Author Response · Authors · 2021-08-06
> **About the entropic scale and the entropy maximization trick: Two fundamental concepts.**
>
> Please, see the Entropy Maximization Trick's fundamental importance in significantly improving the out-of-distribution (OOD) detection performance ([1], Figure 2 and 3; [2], Figure 2), as it extremely and seamlessly increases the entropy of the output of the neural network as recommended by the principle of maximum entropy. Suppose only a single thing should be learned about the proposed solution. In that case, this is the most important point to understand because this is the key to produce competitive state-of-the-art results without adding the complexity typical of current OOD detection solutions.
>
> Additionally, in the same references, please notice the relation between the principle of maximum entropy and the Entropy Maximization Trick. Finally, understand why the Entropy Maximization Trick produces high OOD Detection performance without the side effects of existing approaches, including the Generalized-ODIN (G-ODIN) (see Table 1 of our paper).
>
> Please notice that the authors of [1] tried to combine the entropic scale with traditional SoftMax loss, which is based on affine transformations rather than distances, to perform the entropy maximization trick. However, the results were not consistent. Moreover, they explain [2] that making the entropic scale learnable does not significantly affect the OOD detection performance because they experimentally showed that we are operating in a saturation/plateau region. Some theoretical insights to explain why this is a saturation/plateau region were provided (https://openreview.net/forum?id=f4jw35Vrk6d&noteId=j2wYV1Z0MjG).
>
> Please, notice that even training using the entropic scale, if it is not removed before inference, the entropy does not rise. Consequently, the OOD detection performance also does not increase [1], [2].
>
> [1] Macêdo, D., Ren, T. I., Zanchettin, C., Oliveira, A. L. I., and Ludermir, T. B. "Entropic out-of-distribution detection." *International Joint Conference on Neural Networks (IJCNN), 2021.* preprint: https://arxiv.org/abs/1908.05569.
>
> [2] Macêdo, D., Ren, T. I., Zanchettin, C., Oliveira, A. L. I., and Ludermir, T. B. "Entropic Out-of-Distribution Detection: Seamless Detection of Unknown Examples." *Accepted for publication in IEEE Transactions on Neural Networks and Learning Systems: Special Issue on Deep Learning for Anomaly Detection.* preprint: https://arxiv.org/abs/2006.04005

---

> ### Author Response · Authors · 2021-08-06
> **The isometrization of the distances, the learnable distance scale, and the minimum distance score improved the original IsoMax loss to achieve competitive state-of-the-art performance keeping the solution seamless and working as a SoftMax loss drop-in replacement.**
>
> The major three innovations in the submitted paper that allowed to improve the original IsoMax loss to achieve competitive state-of-the-art performance seamlessly are the isomerization of the distances, the learnable distance scale, and the minimum distance score. The detailed explanations why those changes work are given in the paper.
>
> The experiments are very consistent in showing that those three changes significantly increase the original IsoMax performance in essentially all situations (i.e., combinations of datasets, models, and out-of-distributions). Moreover, the novel version IsoMax-I loss usually overcomes or is extremely competitive with full Mahalanobis, Outlier Exposure, and Generalized-Odin, which is not true when using the original IsoMax loss version. Hence, the submitted paper improved the original IsoMax version to make it achieve competitive state-of-the-art performance.
>
> Besides, the proposal achieves all this without incorporating the existing approaches' special requirements (i.e., hyperparameter tuning and collection of additional data) and undesired side effects (e.g., inefficient inferences and classification accuracy drop). Furthermore, the solution keeps working as a SoftMax loss drop-in replacement for ease of use and large-scale adoption in practice in real-world applications by non-expertise in the field. Finally, the simplicity of the solutions makes us confident that the proposed loss may scale well to large-size images and be applied to other types of data.

---

> ### Author Response · Authors · 2021-08-07
> **About the intense computation post-training phase required by G-ODIN**
>
> After the neural network training finishes, the G-ODIN requires performing inferences on the entire validation set to tune hyperparameters. This operation has to be performed as many times as the number of hyperparameters possible values. Using IsoMax-I, you train only a single model. After the training finishes, you are immediately ready to perform OOD detection. Absolutely no access to a test set containing out-of-distribution data is required to select a specific flavor.

---

> ### Author Response · Authors · 2021-08-13
> **Please, notice we now have results of multiples execution of the same experiment!**
>
> Please, see the post: https://openreview.net/forum?id=f4jw35Vrk6d&noteId=FRsjpDDPKG
>
> Please, see also the comments: https://openreview.net/forum?id=f4jw35Vrk6d&noteId=Qv9bHIGJQRT

---

> ### Author Response · Authors · 2021-08-16
> **Actually, G-ODIN (2020) should have cited the original preprint of IsoMax (2019)**
>
> Dear reviewer,
>
> G-ODIN (2020) was **NOT** the first paper to propose changing the last layer (or, more broadly, the loss) of the neural network to tackle OOD detection!
>
> **The first preprint of IsoMax from 2019 pioneered this type of approach about a year before G-ODIN:**
>
> https://arxiv.org/pdf/1908.05569v1.pdf
>
> Hence, the G-ODIN paper indeed should have cited the first preprint of IsoMax from 2019!
>
> **Regardless of the above observation, G-ODIN proposes a highly complex solution (i.e., it adds a fully connected layer that makes the solution propense to overfitting) that presents too many requirements and drawbacks. Please, see:**
>
> https://openreview.net/forum?id=f4jw35Vrk6d&noteId=GvT-H_VMIEK
>
> **_Inspired by the original IsoMax_, the IsoMax-I improves it without adding new requirements and undesired side effects.**
>
> Please, see how IsoMax-I **SEAMLESSLY** (no additional requirements, no side effects) improves the original IsoMax:
>
> https://openreview.net/forum?id=f4jw35Vrk6d&noteId=2-p24O97S6Q
>
> **Please, see also the link below to notice that IsoMax-I+MDS is completely different from G-ODIN:**
>
> https://openreview.net/forum?id=f4jw35Vrk6d&noteId=BV9NUiVOciM
> ___
> **Albert Einstein: _"Everything Should Be Made as Simple as Possible, But Not Simpler."_**

---

> ### Author Response · Authors · 2021-08-17
> **FINAL REMARKS: A LAST VERY IMPORTANT CLARIFICATION**
>
> Dear all reviewers,
>
> We would like politely to present our final remarks.
>
> As mentioned by the Reviewer gbXF in one post, the maximum entropy principle and the entropy maximization trick are **NOT** the novelty of the current submission, as they were presented in the original IsoMax papers. We never claimed otherwise in the submitted paper.
>
> The novelty of the current submission is the **combination of FIVE necessary and very important modifications** in relation to the original IsoMax+ES solution:
>
> #########################################################################
>
> **1.The normalization of the prototypes.**
>
> **2.The normalization of the features.**
>
> **3.The introduction of the learnable distance scale.**
>
> In our paper, the first three above modifications are collectively called **"the isometrization of the distance"** or **"distance isometrization"**. To the best of our knowledge, this is the first time a paper proposes replacing a distance with what we call **"an isometric distance"**. The insights and motivation to do this are presented here: https://i.imgur.com/TFIqdc3.png
>
> **4.Unlike IsoMax, which initializes the prototypes with zero vectors, we propose initializing the prototypes with random vectors.**
>
> **5.Unlike IsoMax that uses the entropic score, we propose to use the minimum distance score.**
>
> #########################################################################
>
> **By combining these FIVE necessary and very important above modifications (_i.e., if we remove only a single of these, the overall solution fails_)**, we were able to achieve the following:
>
> 1.Unlike IsoMax, **IsoMax-I does NOT present a fatal flaw (i.e., not making in-distribution near prototypes while making out-distribution far from prototypes) from a theoretical point of view**.
>
> Please, see: https://openreview.net/forum?id=f4jw35Vrk6d&noteId=U_sckqmnkp
>
> 2.Unlike IsoMax+ES that presents **medium/regular** OOD detection performance, **IsoMax-I+MDS achieves consistent STATE-OF-THE-ART OOD detection performance, keeping the solution seamless**.
>
> Please, see: https://openreview.net/forum?id=f4jw35Vrk6d&noteId=FRsjpDDPKG
>
> 3.Unlike IsoMax+ES, IsoMax-I produces state-of-the-art results using a **ZERO COMPUTATIONAL COST SCORE (i.e., the minimum distance score)**.
>
> ___
>
> **To the best of our knowledge, the proposed approach is the only currently available solution that SIMULTANEOUSLY is SEAMLESS (no hyperparameters tuning, no additional data collection, no classification accuracy drop, no inefficient inferences) and presents ROBUST STATE-OF-THE-ART OOD detection performance.**
>
> We sincerely believe this is a NOVEL and RELEVANT contribution for our community.
>
> ___
>
> For references and details, please see:
>
> https://openreview.net/forum?id=f4jw35Vrk6d&noteId=2-p24O97S6Q
>
> ___
> **Albert Einstein: _"Everything Should Be Made as Simple as Possible, But Not Simpler."_**

---

### Official Review · Reviewer_gbXF · 2021-07-16

**Rating:** 4
**Confidence:** 5

**Summary:**

The authors propose some marginal modifications to the IsoMax loss (i.e. normalising the logits and the prototypes, multiplying the distance between the logits and the prototypes by a non-negative scalar, using a different uncertainty score based on the distance from the prototypes).

Positive aspects:
- the formulation makes sense, and the distance score is simple but reasonable

Unclear points:
- why is it necessary to initialize the prototypes randomly? Other papers like [1] show one can select the prototypes to make the training procedure stable.
- why should multiplying by a fixed scalar |d_s| (learned only at training time, but not depending on the input) have any effect on the performance?
Weaknesses:
- Novelty is marginal
- Experiments are performed with only one seed (need to repeat the experiments with at least 5 different seeds)
- Does not compare against baselines with similar concepts [1, 2], and makes unfair comparisons with methods that leverage more information/have accuracy drops; recent literature has shown there are many ways to preserve accuracy and improve OOD detection [3,4,5] that would make fair comparison baselines
- Uncertainty estimation is more than just out of distribution detection: it would be good for the authors to show calibration performance, and evaluate accuracy and calibration on corrupted datasets (e.g. CIFAR-10/100-C) [6].
- The clustering properties of the proposed method are not analysed in depth
- Given the few and statistically unreliable experiments presented, at least showing the method could scale to ImageNet would have been interesting

[1] Class Anchor Clustering: a Loss for Distance-based Open Set Recognition, Dimity Miller, Niko Sünderhauf, Michael Milford, Feras Dayoub
[2] Uncertainty Estimation Using a Single Deep Deterministic Neural Network, Joost van Amersfoort, Lewis Smith, Yee Whye Teh, Yarin Gal
[3]Training independent subnetworks for robust prediction, Marton Havasi, Rodolphe Jenatton, Stanislav Fort, Jeremiah Zhe Liu, Jasper Snoek, Balaji Lakshminarayanan, Andrew Mingbo Dai, Dustin Tran
[4] BatchEnsemble: An Alternative Approach to Efficient Ensemble and Lifelong Learning, Yeming Wen, Dustin Tran, Jimmy Ba
[5] Fast Predictive Uncertainty for Classification with Bayesian Deep Networks, Marius Hobbhahn, Agustinus Kristiadi, Philipp Hennig
[6] Benchmarking Neural Network Robustness to Common Corruptions and Perturbations, Dan Hendrycks, Thomas Dietterich

**Limitations And Societal Impact:**

Yes

**Main Review:**

Originality: the paper is incremental and does not introduce particularly innovative concepts or methodologies. It just slightly modifies the IsoMax technique.

Clarity: the paper is clearly written and organised. The extensive presence of allegedly unfair comparison does not make much sense to me, especially considering the experiments have been performed without averaging over (at least) 5 seeds, and many times the differences in these cases marginal or the method is underperforming. There is also no reason to break the presentation of the result for one type of experiments in so many tables.

Significance: The paper does not show statistically significant results, and the results seem to be mixed in many cases, also based on which metric is used. The choice of the baselines is questionable and neglects most of the recent literature (I cited just a few of the most recent techniques).

Quality: the paper is well written, but the experiments are not statistically significant nor extensive enough for acceptance.

**Time Spent Reviewing:**

3

---

> ### Author Response · Authors · 2021-08-05
> **The experimental results are indeed highly consistent across many models, datasets, and out-of-distributions. This is remarkable considering that our approach does not involve hyperparameters tuning and avoids common drawbacks of current approaches.**
>
> Unfortunately, we had not enough time to run many experiments before the submission deadline. Nevertheless, we run many executions during the review process, and the results are essentially the same presented in the paper. If the reviewers desire, we may update the final version of the paper to include these multi-execution results. Considering we made the code available, the reviewers may check that the results are consistent when we execute the experiments many times.
>
> **Even considering only the results currently in the paper as initially submitted, the results are extremely compelling. We used many datasets, models, and out-of-distributions, and the margins are usually significantly safe in all distinct scenarios in the fair comparison tables (Tables 2 and 3). The margins in favor of the IsoMax-I loss are smaller when we unfairly compare with full Mahalanobis (Table 4), which requires hyperparameters tuning, adversarial examples, feature extraction, and training additional models. Moreover, in the unfair comparison against outlier exposure (Table 5), which needs additional data while the Isometric IsoMax (IsoMax-I) loss does not, the margins in favor of the IsoMax-I loss are substantial in the CIFAR100, which is a very important case since we have few labels per class and many classes.**
>
> The paper [1] is very recent and has been published in the IEEE Explorer of the *2021 IEEE Winter Conference on Applications of Computer Vision (WACV)* on June 14, after the submission deadline (https://ieeexplore.ieee.org/document/9423243). Unfortunately, we cannot compare against the results of the mentioned paper, as they do not use the traditional Dan Hendrycks et al. benchmark(datasets, models, and metrics) used by the ODIN, Mahalanobis Outlier Exposure, Generalized ODIN, and other classic out-of-distribution detection approaches. Unlike in the Dan Hendrycks et al. benchmark (which uses different datasets for in-distribution and out-of-distribution), in the benchmark used in [1] (we call it the "Open Set Recognition" benchmark), untrained classes of the same in-distribution datasets are used as out-of-distribution, making a comparison of IsoMax-I loss with the approach in [1] not possible right now.
>
> **Rather than clustering and uncertainty experiments, similar to the seminal classical OOD detection papers mentioned, we focus our study on the out-of-distribution detection problem. Additionally, in the title or abstract, we never claimed that our paper deals with anything other than the already challenging enough out-of-distribution detection problem. We indeed compared with the major OOD detection approaches (the scope of the paper presented in the title and the abstract) that follow the Dan Hendricks et al. benchmark (e.g., ODIN, Mahalanobis, and Outlier Exposure).**
>
> ***Moreover, most of the approaches the reviewer mentioned appear to focus on uncertainty estimation (rather than OOD detection, which is the scope of our paper) or present slow and energy-inefficient inference inefficient, as they are usually based on Bayesian Deep Networks or ensemble approaches. Additionally, they have many other undesired collateral effects not present in our approach, which is also much easier to use. Please notice that nothing prevents us from applying all the mentioned Bayesian, ensemble, and uncertainty estimation techniques using IsoMax-I loss rather than the SoftMax loss to start from a much higher baseline performance.***
>
> Regarding randomly initializing weights, allowing this is actually an advantage compared with requiring specialized initialization procedures. Additionally, we can see Figure 7 of the paper *"Entropic Out-of-Distribution Detection: Seamless Detection of Unknown Examples." Accepted for publication in IEEE Transactions on Neural Networks and Learning Systems (IEEE TNNLS): Special Issue on Deep Learning for Anomaly Detection* (https://arxiv.org/pdf/2006.04005.pdf) that IsoMax loss training is extremely stable without requiring specialized initialization.
>
> We observe that the approach proposed in [1] requires two hyperparameters (alpha and lambda) to be tuned, while the IsoMax-I loss does not require hyperparameter tuning. There is no magic: if you have hyperparameters to tune similar to G-ODIN, some training data is lost to be used for validation. Consequently, we train the neural network with fewer data, and classification accuracy drop is inevitable. This is more significant in the important case of few labeled data per class (e.g., CIFAR100). The IEEE TNNLS above-mentioned paper (Figure 3) (https://arxiv.org/abs/2006.04005) shows that IsoMax loss does not present classification accuracy drop even when subjected to an extremely low amount of labeled training data per class. Hence, we suggest testing the solution proposed in [1] using CIFAR100 with only 100 labeled training images per class as in (https://arxiv.org/abs/2006.04005, Figure 3).
>
> Moreover, the approach in [1], unlike IsoMax-I loss, has non-trainable (anchored) prototypes. Hence, we do not think this approach produces promising generalization power to data outside the ones used in the benchmark. Finally, IsoMax-I loss is much easier to use than [1], as the former works as a simple SoftMax loss drop-in replacement. IsoMax-I loss can be used by changing two lines of PyTorch code. We need state-of-the-art, easy-to-use approaches to be published to popularize the adoption of out-of-distribution detection in real-world applications, mainly for people who are not specialists in the area.
> ___
> ***Final remarks: Considering the reviewers did not find anything technically wrong with our approach (actually, the opposite is true, as the majority saw a robust proposal with competitive state-of-the-art performance), we suggest accepting the paper and let people decide by themselves which one (IsoMax-I loss or an existent approach) they prefer to use in practice in the real world.***

---

> > ### Comment · Reviewer_gbXF · 2021-08-11
> > **Lack of novelty can be fine if extensive experimental evidence is provided**
> >
> > Significant margins are not a replacement for multi-seed experiments: a method can have high variance in the performance and a lucky seed might wrongly suggest a method is superior to another. Multi-seed experiments have to be provided to give confidence in the performance of the method.
> >
> > I don't see any technical limitation in using [1] as a baseline, or any other of the recent competing methods that assume no knowledge of the OOD data (e.g. DUQ, SNGP, Batch-Ensembles, Deep Ensembles, Snapshot Ensembles etc.). Regarding the comparison with [1], although it is not strictly necessary, one cannot just make hypotheses about its performance. Your statements should be supported by experiments.
> >
> > If your paper had significant novelty with respect to previous works, limiting the scope to OOD detection could have been acceptable. However, the paper is incremental and the novelty is extremely marginal: other kinds of experiments evaluating the performance of the uncertainties produced by your method or analyses providing insights are strongly recommended (see previous response).
> >
> > I agree your method is very simple to implement, and it's a strength of it. The main problem is the lack of novelty combined with the lack of extensive experimental evidence.

---

> > > ### Author Response · Authors · 2021-08-11
> > > **How the only available seamless robust state-of-the-art performance OOD detection approach may lack novelty?**
> > >
> > > Regarding novelty, please see the discussion in this post:
> > >
> > > https://openreview.net/forum?id=f4jw35Vrk6d&noteId=AigRyoC75-R
> > >
> > > ***How the only currently available seamless robust state-of-the-art performance OOD detection approach may lack novelty?***
> > >
> > > Or should we consider novelty approaches that require/produce: many hyperparameters, additional data, inefficient inferences, classification accuracy drop, are not robust for all situations, requires fine-tuned initialization, present no learnable prototypes?
> > >
> > > ***In our approach, we start from an entirely novel mindset:  the entropy of the in-distribution data decreases naturally during training because we are minimizing the loss. However, as recommended by the principle of maximum entropy, it is done as slowly as possible to avoid overconfidence.***
> > >
> > > ___
> > >
> > > Question 1: Considering you say that what we did is not novel, please cite ANY other approach that keeps the entropy of the in-distribution as high as possible rather than trying to minimize it to make it more far apart from OOD examples?
> > > ___
> > >
> > > Question 2: Considering you say that what we did is not novel, why are you SURPRISED by the fact that we are keeping the entropy of in-distribution data as high as possible?
> > > ___
> > >
> > > Question 3: Considering you say that what we did is not novel, please, cite ANY other OOD detection approach that is SEAMLESS in the terms we defined in our paper? Please, notice that our approach is seamless while providing state-of-the-art performance.
> > > ___
> > >
> > > Question 4: Considering you say that what we did is not novel, why do you keep asking the reasons why we maximize the entropy of the in-distribution?
> > > ___
> > >
> > > Question 5: Please, do you really notice that in Tables 4 and 5 our approach is outperforming (or at least being competitive with) state-of-the-art approaches even operating under EXTREMELY MORE RESTRICTIVE CONDITIONS? We believe you should consider this extremely remarkable!
> > > ___
> > >
> > > We have about two and half pages showing results of experiments. We used many different models, datasets, and out-of-distribution data. The results are consistent for many distinct datasets, models, and out-of-distribution.
> > >
> > > Please, notice that running using too many models, datasets, and out-of-distribution data is much more important than running the same experiment many times, as the difference in results are very tiny in the latter, while the difference in performance may differ much more substantially in the former.
> > >
> > > Notice that using a broad set of datasets, models, and out-of-distribution provides much more evidence of generalization power than running the same experiment many times!
> > > ___
> > >
> > > Question 6: Do you sincerely believe that the results in Tables 2, 3, 4, and 5 could simultaneously happen by chance considering that we used a vast amount of different datasets, models, and out-distributions?
> > > ___
> > >
> > > Please, run the code yourself and see that our state-of-the-art results are remarkably consistent.
> > >
> > > The final paper will have mean and standard deviations.

---

> > > > ### Comment · Reviewer_gbXF · 2021-08-11
> > > > **Enough talking, provide theorems or experiments**
> > > >
> > > > The results are not state-of-the-art because multiple-seed experiments are missing. Even in my own experiments for other projects I could obtain seeds with 99% AUROC/AUPR in OOD, but on average the performance decreased (because the method had high variance). It is not my responsibility to run your code and do experiments following appropriate scientific procedures, it is yours.
> > > >
> > > > Also, **you are still ignoring my recommendation to run more experiments (many of which do not require to retrain the models) and I don't think I'll waste more time trying to convince you that more experiments can make your submission better.**
> > > >
> > > > As for the novelty, the discussion is becoming ridiculous. I have cited works proving that uncertainty should increase getting far away from the training data, it is your responsibility to argue against those works and prove that your method should theoretically obtain better results even if the maximum entorpy principle is applied on all points. I agree that in your model the network will try to decrease the entropy in order to preserve the accuracy on in-domain samples, but it needs further elaboration to be a worthy theoretical contribution. Furthermore, what you point out is not novel at all, because IsoMax already applies the trick you use. You only modify IsoMax slightly. There are already 2 papers published using Isomax, with very similar contents. Your paper adds very little to that. For the paper to be accepted 2 factors are considered: either the novelty or the experimental quality. Both are missing in this submission. Experiments should provide either deep insights in the understanding of the model's behaviour and why it succeeds where other models fail, or it should show the model is versatile enough to perform other tasks. You do not provide any of those things. In its current state, this can be a quick workshop paper, not a full conference paper.  I am not saying that your method is bad, I am saying you should work on improving the submission.
> > > >
> > > > **I've very gently suggested many times now that given the little novelty you could provide more experiments and insights to improve the submission and make it acceptable. I also pointed at experiments that can be done without re-training.**
> > > >
> > > > In conclusion, **given the tone of the responses I'm receiving from the authors, and the fact my request for further experiments has not been responded to, I think I've dedicated enough time to the discussion. I will only respond or change my evaluation (if appropriate) if at least one of the following conditions is satisfied**:
> > > >
> > > > 1. multiple seed experimental results are provided with means and standard deviations for all tables and all methods;
> > > >
> > > > 2. calibration metrics, reliability plots and misclassification detection experiments are carried out and the results are appropriately reported (using adequate metrics);
> > > >
> > > > 3. A series of theorems/a proper theoretical discussion proving the method is theoretically grounded for OOD detection and that the theoretical works of others are wrong (see the SNGP paper I cited).
> > > >
> > > > 3. (optionally) data-shift experiments are performed (measuring accuracy and calibration).
> > > >
> > > > As a final suggestion to the authors: for future rebuttals, keep a moderate tone and gently answer to the questions synthetically and effectively. Reviewers spend their time (for free) evaluating your method and try as much as possible to improve your submission.

---

> > > > > ### Author Response · Authors · 2021-08-11
> > > > > **We already have the results of many experiments. We are collection it to present them.**
> > > > >
> > > > > As we said, we already performed the experiments many times after submission.
> > > > >
> > > > > **We are not ignoring the recommendation of adding multiple runs to the paper. However, we did not know we could show them to you during the review period.**
> > > > >
> > > > > Considering we are asking to see them, we are preparing them to show to you.
> > > > >
> > > > > This is the fastest way we can to achieve one of your condition to improve the score. This is the condition 1 and your main complaint. We may take some time because we have many tables.
> > > > >
> > > > > We will provide follow up to this discussion.

---

> > > > > > ### Comment · Area_Chair_ji6Z · 2021-08-11
> > > > > > **You're welcome to post new experimental results requested by the reviewers**
> > > > > >
> > > > > > You can either post numbers directly in the comments here, or – if you are careful to maintain anonymity – you may link to a figure posted on an anonymous outside site, e.g. imgur.com.
> > > > > >
> > > > > > This discussion has gotten rather heated; I'd like to suggest that it would be most useful for all parties to stick to brief responses to specific questions in your replies.

---

> > > > > > > ### Author Response · Authors · 2021-08-13
> > > > > > > **We posted the new experimental results on the front page to make them more visible**
> > > > > > >
> > > > > > > We preferred to post the new experimental results on the front page to make them more visible.

---

> > > > > ### Author Response · Authors · 2021-08-12
> > > > > **A MATTER OF FAIRNESS**
> > > > >
> > > > > We politely ask the reviewer to notice that presenting uncertainty estimation or domain shift results is neither usual nor mandatory for an **out-of-distribution detection focused paper**. Please, see the ODIN, Mahalanobis, and Gram matrices papers.
> > > > >
> > > > > For a more recent example, we present the Energy-based approach published last year in NeurIPS. We are indeed showing **MANY MORE** tables and results than the mentioned paper. Additionally, like many other papers focused on OOD detection, the mentioned paper also does **NOT** present uncertainty estimation or domain shift results, as OOD detection is already a hard enough task to deal with. We never claimed uncertainty estimation or domain shift was in the scope of our study in the title, abstract or paper.
> > > > >
> > > > > Similar to what we are doing, they compare OOD detection performance among the most common and largely used OOD detection approaches. Other papers may investigate questions regarding how appropriate our approach is when dealing with uncertainty estimation or domain shift.
> > > > >
> > > > > Finally, the mentioned paper also **DOES NOT** show results using mean and variation. **Nevertheless, as this is the reviewer's main concern, and it is indeed a fair criticism, we are working to provide them for all FIVE tables that contain results.**

---

> > > > > ### Author Response · Authors · 2021-08-12
> > > > > **A necessary clarification**
> > > > >
> > > > > The reviewer wrote:
> > > > >
> > > > > _As for the novelty, the discussion is becoming ridiculous. I have cited works proving that uncertainty should increase getting far away from the training data, it is your responsibility to argue against those works and prove that your method should theoretically obtain better results even if the maximum entropy principle is applied on all points._
> > > > >
> > > > > We never said that uncertainty should not increase getting far away from the training data. We also never said that we applied the maximum entropy principle to all points. Actually, we cannot even try to maximize the entropy in out-of-distribution data, as we do not have access to them during training. We said that we apply the maximum entropy principle to the entire **training data**.
> > > > >
> > > > > Please, see Figure 1. Using IsoMax-I, the out-of-distribution data is indeed becoming far from in-distribution and prototypes during inference, which is precisely the desired behavior you mentioned. **This is precisely the innovation of IsoMax-I: this desired behavior is achieved using an entirely novel approach!** The desired behavior you mentioned is obtained simply by applying the maximum entropy principle on training data.
> > > > >
> > > > > ##############################################################################
> > > > >
> > > > > ***This makes out-of-distribution data become more far away from prototypes than in-distribution, and this is all we need to perform OOD detection! It appears that when you try too hard to get in-distribution too close to you (just like SoftMax and the usual OOD detection approaches does), the out-of-distribution data also comes fast in your direction, making OOD detection hard!***
> > > > >
> > > > > ##############################################################################
> > > > >
> > > > > **Hence, producing the maximum entropy possible on training (in-distribution data) makes out-of-distribution examples become even farther away from the prototypes than in-distribution data (Figure 1)! We understand that this may be not intuitive and even surprising, and this is exactly the reason why the approach is truly original.**
> > > > >
> > > > > We all agree that we need out-of-distribution data more distant from prototypes than in-distribution data. The question is **HOW** we get this desired behavior. The authors propose to get this desired behavior by maximizing the entropy of the probability distribution. The entropy maximization trick is how they achieve this.
> > > > >
> > > > > **Regarding the specific innovation of the IsoMax-I, please pay attention to Figure 1. The behavior you mention that we should try to obtain (i.e., getting in-distribution closer to prototypes and getting out-of-distribution far away from prototypes) is precisely what IsoMax-I achieves, while the IsoMax does not! The original IsoMax presented high OOD detection performance using the entropy of the output, while the IsoMax-I uses a simple distance score to significantly improve the OOD detection performance and safely outperform the original IsoMax!**
> > > > >
> > > > > This is possible exactly because IsoMax-I is clever enough to get in-distribution closer to prototypes and get out-of-distribution far from prototypes (even without access to OOD samples during training!), which is exactly the behavior you said we should try to obtain. Hence, all paper you mentioned are actually showing that the proposed approach makes sense. However, we obtain this desired behavior using an entirely novel and somewhat not intuitive manner. This is where the novelty resides.
> > > > >
> > > > > **Therefore, Figure 1 shows that the isomerization of the distance, the learnable distance, the normalization of the prototypes and weights, and the minimum distance score (all this from the IsoMax-I proposed paper) contributed decisively to this: obtain seamless robust state-of-the-art OOD detection results. These procedures significantly improved the original IsoMax OOD detection performance.**

---

> ### Author Response · Authors · 2021-08-06
> **Why do we need the entropic scale and the entropy maximization trick?**
>
> The reviewer wrote:
>
> _"why should multiplying by a fixed scalar |d_s| (learned only at training time, but not depending on the input) have any effect on the performance?"_
>
> The entropy maximization trick (i.e., training with the entropic scale but removing it for inference) is a highly innovative and easy way to produce extremely HIGH entropy posterior probability distributions. Why is this important? Because the principle of maximum entropy states/affirms that high entropy probability distributions are the best possible option to produce more realistic predictions and reduce bias (this is why the approach is called ENTROPIC Out-of-Distribution Detection).
>
> The original ENTROPIC Out-of-Distribution Detection papers (https://arxiv.org/pdf/1908.05569.pdf, Figure 2 and 3 and https://arxiv.org/pdf/2006.04005.pdf, Figure 2) present an extensive and detailed explanation about why producing neural networks with outputs with high entropy probability distributions produces high OOD detection performance. Considering that the universally used SoftMax loss produces outputs with extremely LOW entropy posterior probability distributions, the same maximum entropy principle explains why SoftMax loss produces extremely low OOD detection performance.
>
> This is the central point and fundamental principle the entire approach is based on, allowing IsoMax loss variants to work extremely well without producing undesired side effects. Most people do not even pay attention to this foundational aspect of the work. They usually only pay attention to the distance-based aspect of the solution. However, the mentioned original papers show that PRODUCING HIGH ENTROPY OUTPUT DISTRIBUTIONS IS EVEN MORE IMPORTANT THAN USING A DISTANCE IN THE LOSS AND THAT THE ENTROPY MAXIMIZATION TRICK IS A SEAMLESS AND STRAIGHTFORWARD WAY TO ACHIEVE THIS.
>
> Thanks for making this question!

---

> > ### Comment · Reviewer_gbXF · 2021-08-11
> > **You should be more precise**
> >
> > I didn't ask why maximising the entropy is important, I asked why the entropy maximisiation trick is implemented by multiplying by a fixed scalar |d_s| (learned only at training time, but not depending on the input).
> >
> > Furthermore, the fact the entropy is increased regardless which input is chosen can be harmful. For how OOD tasks are evaluated it is not true that producing always high entropy outputs implies better OOD performance. For the evaluation metrics to be maximised one has to separate as much as possible the entropy distributions of in-distribution and OOD samples, so the goal should be to have low entropy on in-distribution samples and high entropy on out-of-distribution samples. Indeed, the AUROC/AUPR metrics assume a thresholding classifier.
> >
> > Additionally, producing high entropy outputs regardless of what the input is (in-distribution or OOD) can yield to the following issues: 1) reduced calibration (the network becomes underconfident for no reason; the model should be "appropriately" confident, underconfidence is not a fix for overconfidence), 2) decreased mis-classification rejection performance (due to the underconfidence, the network cannot reject samples on which it is wrong). The network might also show reduced performance on data-shift. If you propose a method that increases entropies over every input, then you must prove that the effect it has on the uncertainty does not damage these other tasks, or at least acknowledge that as a limitation.

---

> > > ### Author Response · Authors · 2021-08-11
> > > **Please, could you clarify your question?**
> > >
> > > The reviewer wrote:
> > >
> > > _"I didn't ask why maximising the entropy is important, I asked *why* the entropy maximisiation trick is implemented by multiplying by a fixed scalar |d_s| (learned only at training time, but not depending on the input)."_
> > >
> > > What do you mean by _"fixed scalar |d_s|"_? The fixed scalar we have is called **the entropic scale E_s, not |d_s|**. The |d_s| is **the distance scale, which is a learnable scalar**.
> > >
> > > Additionally, what do you mean by _"learned only at training time, but not depending on the input"_. Do you know something that may be learnable at inference time? How something that is learnable does not depend on the input? If something is learnable, naturally the data we use during training affects its learning.

---

> > > > ### Comment · Reviewer_gbXF · 2021-08-11
> > > > **Respond in a pertinent way and use adequate tones**
> > > >
> > > > First, the tone of your response sounds quite inappropriate. I would remind the authors to respond adequately, as I will not tolerate inadequate tones.
> > > > Second, you did not ask for clarifications in your first response.
> > > > Third, the lack of precision was referred to the entropy maximisation discourse, that you deliberately ignored in your response. I invite you to provide an extensive response about that.
> > > > Fourth, here are the clarifications. The constant is learned at training time but is not a function of the test input. Why the distance scale should not depend on the tested input?
> > > > Furthermore, hyperparameters (like constants, in your case) can of course be tuned post-hoc (after the training ends). For instance, one can compute the Hessian of a Laplace Approximation over the output after the training ends, and tune its priors based on specific criteria.

---

> > > > > ### Author Response · Authors · 2021-08-11
> > > > > **Additional explanations asked by the reviewer.**
> > > > >
> > > > > The reviewer wrote:
> > > > >
> > > > > _Third, the lack of precision was referred to the entropy maximisation discourse, that you deliberately ignored in your response. I invite you to provide an extensive response about that._
> > > > >
> > > > > We did not "deliberately ignored" the "entropy maximisation discourse". We believe we already explained it. However, considering we were invited by the reviewer, we will try to provide an even more "extensive response" about that. We really want to convince you about the merits of our work.
> > > > >
> > > > > We said that it is reasonable to think that we should impose low entropy for in-distribution while forcing high entropy for out-distribution. This is actually what all approaches known so far try to do. This is also naturally what the reviewer keep saying we should have done. We actually tried such approaches too many times before designing IsoMax-I. Intuitively, we think: to separate in-distribution from out-distribution, we should impose low (or at least not so high) entropy to in-distribution and high entropy to out-distribution.
> > > > >
> > > > > It is an entirely reasonable approach to follow. This is why ALL approaches so far have tried different ways of doing exactly this.
> > > > >
> > > > > For example, the Outlier Exposure forces high entropy in additional data while allowing the SoftMax free to impose low entropy for in-distribution examples (this is what SoftMax always does: imposing low entropy for in-distribution data).
> > > > >
> > > > > However, the hard truth is the following: They (the current approaches) all failure in delivering a SEAMLESS ROBUST STATE-OF-THE-ART SOLUTION for the OOD Detection problem. We followed an entirely new path!
> > > > >
> > > > > We will blindly follow what the principle of maximum entropy (which is related to the second law of thermodynamics) tells us to do: We will construct probability distributions with the maximum entropy possible. Do we try to maximize the entropy of in-distributions? Yes! We do EXACTLY this. Does this make any sense? The principle of maximum entropy states that it indeed makes sense. Even better: the experiments agree! By maximizing the entropy of our distributions, we delivered the ONLY SEAMLESS ROBUST STATE-OF-THE-ART SOLUTION FOR THE OOD DETECTION PROBLEM.
> > > > >
> > > > > But how is this possible? Answer: By following the principle of maximum entropy (i.e., maximizing the entropy of the in-distribution), we observe that (Please, see Figure 1 of our paper), it becomes straightforward to separate in-distribution from out-distribution examples. **In such cases, the out-distribution examples present EVEN HIGHER entropy than the in-distribution examples, making detection OOD examples trivial! It may be surprising!** It may not be intuitive! But the experiments show that this indeed works very well!
> > > > >
> > > > > But what about the out-distribution? We do not care about it. We have no access to it during training. We just believe when the maximum entropy principle says that: "THE BEST POSSIBLE PROBABILITY DISTRIBUTION (I.E., THE ONE WITH THE LEAST BIAS) IS THE ONE THE PRESENTS THE MAXIMUM ENTROPY." In other words, you should do the least amount of assumptions about the data possible. The other posts add more information to this.
> > > > >
> > > > > In the terms of the principle of maximum entropy, you should be as underconfident as possible, which is exactly the opposite of our current neural networks! This is the BEST possible probability distribution considering the data you have available!
> > > > >
> > > > > But what about uncertainty?
> > > > >
> > > > > First of all: In the title, the abstract, and the paper, we simply claim this: we propose a seamless robust state-of-the-art solution for the OOD DETECTION PROBLEM following an ENTIRELY NOVEL DIFFERENT APPROACH.
> > > > >
> > > > > As OOD detection and uncertainty estimation are strongly related task, we sincerely believe that IsoMax-I will performance much better than the extremely overconfident SoftMax, mainly after we apply the many available uncertainty estimation to it. We remember that we never claimed in the title, the abstract, or in the paper nothing regarding uncertainty estimation.
> > > > >
> > > > > We were always very clear that we are focused, just like the classical OOD detection paper (e.g., ODIN, Mahalanobis, Outlier Exposure), on the already challenging enough OOD detection problem.
> > > > >
> > > > > Just like you may be surprised that maximizing the entropy of the in-distribution produces SEAMLESS ROBUST OOD DETECTION PERFORMANCE, we may be surprised that IsoMax-I actually presents extremely high quality uncertainty results.
> > > > >
> > > > > If in the future, without precipitated and unfair speculations, we realize that IsoMax-I is does not present much better uncertainty performance than SoftMax even after adding to it modern uncertainty estimation techniques, there is no problem. Everyone that wants a trivially easy to use robust seamless state-of-the-art OOD detection approach may still use IsoMax-I in case they do not care about uncertainty estimations.

---

> > > > ### Author Response · Authors · 2021-08-11
> > > > **Further elaborating on the answer we gave to the reviewer.**
> > > >
> > > > We believe the reviewer mean: Why does multiplying the logits by a fixed scale (the entropic scale) during training but remove it before inference increases the entropy, as recommended by the principle of maximum entropy?
> > > >
> > > > We will give theoretical insights, experimental evidence, and also refer the reviewer to the original IsoMax paper for further reading.
> > > >
> > > > First, the theoretical insights. The cross entropy minimization tends to force the neural network to produce extremely overconfident predictions. This happens because it tries to maximize the logits of each class. During inference, the network gives a very high value for one of the logits and very low value for the others logits. This produces overconfidence and extremely low entropy.
> > > >
> > > > Here comes the entropic scale. When we multiply the logits DURING training by a high value, considering the logits are inside an exponential function, it is possible to allow the cross entropy loss to go to almost zero (i.e., train the neural network) WITHOUT PRODUCING EXTREMELY HIGH LOGITS and, consequently, extremely high overconfident predictions during inference!
> > > >
> > > > Please, notice that the entropic scale used during training need to be removed before inference, OTHERWISE WE ARE DOING NOTHING! If we do not remove the entropic scale for performing inference, everything would behave just like having using them during training! In other words, if we kept the logits multiplied by the entropic scale during inference, we are actually using exactly the same very high logits produced by the cross entropy loss!
> > > >
> > > > Please, see a more detailed explanation in Section III, A of https://arxiv.org/pdf/2006.04005.pdf.

---

> > > > > ### Comment · Reviewer_gbXF · 2021-08-11
> > > > > **Please, answer the question about entropy**
> > > > >
> > > > > I understand how the entropy trick works and I have been imprecise in saying that |d_s| implements it. My question (not exactly my main question) is why the inclusion of |d_s| (which is independent of the input) should affect the output and how.
> > > > > I remark this is the minor of my concerns, and honestly can go completely unanswered.
> > > > >
> > > > > Instead, my biggest concern (for which I gave a thorough discussion) is about increasing the entropy everywhere, and it has not been addressed at all. I pointed out that increasing entropy everywhere can harm the uncertainty quality produced by the neural network (including OOD detection performance). Empirical evidence should be provided about it. Calibration metrics and misclassification rejection results should be provided. Also the evaluation under data-shift is recommended. The execution of these experiments does not require re-training any model, it only requires to evaluate the models on test sets. Hence it is effortless and their lack is a significant limitation of the work, especially considering its incremental nature.

---

> > > > > > ### Author Response · Authors · 2021-08-11
> > > > > > **Clarifications about the principle of maximum entropy.**
> > > > > >
> > > > > > The reviewer wrote:
> > > > > >
> > > > > > _"my biggest concern (for which I gave a thorough discussion) is about increasing the entropy everywhere, and it has not been addressed at all."_
> > > > > >
> > > > > > We need to "increasing the entropy everywhere" because this exactly what the principle of the maximum entropy tell us to do and because the results show that this indeed work! The principle is very clear is saying that you should begin the most uncertainly about everything and, AS SLOWLY AS POSSIBLE, start to updating your beliefs.
> > > > > >
> > > > > > We follow exactly what the principled tell us to do. We increased the entropy of the entire distributions. This allowed us to present the only available seamless OOD detection approach currently available. Even more, it is robust and present state-of-the-art performance.
> > > > > >
> > > > > > Other approaches try to make the entropy of the in-distributions low, and what happens? They all failed to construct a SEAMLESS ROBUST STATE-OF-THE-ART OOD SOLUTION. They ALWAYS produce undesired side effects (e.g., ODIN, Mahalanobis, Outlier Exposure, Generalized ODIN). This is the whole truth, no matter how hard you try not to believe in this fact.

---

> > > ### Author Response · Authors · 2021-08-11
> > > **Complementing the response for the reviwer's comments**
> > >
> > > It indeed makes sense to initially try the approach mentioned by the reviewer. We may initially think that we should try as much as possible to make the in-distribution to have low entropy and out-distribution high entropy. This way of thinking makes sense. This is exactly why all previous/current OOD detection approaches follows this path by trying to make the maximum possible separation between in-distribution and out-of-distribution. We ourselves tried too many times OOD approaches following this mindset. We always failed to obtain a SEAMLESS STATE-OF-THE-ART out-of-distribution solution using this reasonable way of thing.
> > >
> > > **However, this is not what the Principle of Maximum Entropy tells us to do!**
> > >
> > > We must believe more in principles than in our fallible initial intuitions. The Principle of Maximum Entropy is EXTREMELY correlated to the BAYES THEOREM and, more broadly, BAYESIANISM in general. It allows us to construct and explain all probability distribution we use in statistics. All them are based on this principle (https://en.wikipedia.org/wiki/Principle_of_maximum_entropy). It is in the cornerstone of the Bayesian statistics. It is even the principle that establishes the destiny of the entire universe. Please see the references [2], [3], [4], and [5], all cited and discussed in [1] to a discussion about the principle of the maximum entropy. The approach is not called *Entropic* Out-of-Distribution Detection by chance.
> > >
> > > We need to begin with total doubt about everything! After adding evidence, we need to, as slowly as possible, to update our trust in something. The Principle of Maximum Entropy says that the BEST PROBABILITY DISTRIBUTION POSSIBLE, THE ONE WITH THE LOWEST BIAS POSSIBLE, is the one the produces maximum entropy, _REGARDLESS OF WHEN YOU ARE ACTUALLY RIGHT_. For the cited principle, we should be underconfident even when you a right, _BECAUSE YOU HAVE NOT ENOUGH EVIDENCE YET TO BE SO SURE ABOUT THAT DECISION, EVEN BEING INDEED RIGHT IN THIS PARTICULAR CASE!_.
> > >
> > > Therefore, the entropy maximization trick is a clever way to obey the Principle of Maximum Entropy by producing probabilities distributions with the maximum possible entropy/uncertainty. We know that by following the Principle of Maximum Entropy, we make our in-distribution present high entropy (low confidence).
> > >
> > > **But guess what? The out-of-distribution presents even higher entropy and even lower confidence, and this is all we need to separate between in-distribution and out-distribution. We all now have to recognize that it allowed us to construct the ONLY SEAMLESS STATE-OF-THE-ART PERFORMANCE OOD detection approach.**
> > >
> > > Regarding possible negative impacts in terms of being "appropriately" confident, you are being extremely speculative about saying that our approach will not produce high quality uncertainty performance, as we have no experiments to affirm this. Considering the extreme success dealing with the OOD detection problem and that uncertainty estimation is a strongly correlated problem, we cautiously believe that applying the usual uncertainty estimation techniques to our approach will make it provide much better results than using the SoftMax loss.
> > >
> > > Moreover, as mentioned in the title and abstract, our intention is to propose an out-of-distribution detection approach. Additionally, what exactly is "appropriately" confident when we are dealing with OPEN SET PROBLEMS in which we do not know the proportion of in-distribution examples in relation to out-of-distribution examples we will observe in the field?
> > >
> > > Maybe trying to produce "appropriately" confident solutions is at odds with producing the best OOD detection solution or the creation of the least biased possible probability distributions (please notice that the principle of maximum entropy assured us that we are  producing the least biased probability distribution possible). Accepting the paper and making more studies about this is the only way to know for sure.
> > >
> > > ___
> > >
> > > [1] Macêdo, D., Ren, T. I., Zanchettin, C., Oliveira, A. L. I., and Ludermir, T. B. "Entropic Out-of-Distribution Detection: Seamless Detection of Unknown Examples." Accepted for publication in IEEE Transactions on Neural Networks and Learning Systems: Special Issue on Deep Learning for Anomaly Detection. preprint: https://arxiv.org/abs/2006.04005
> > >
> > > [2] J. Pearl, “Probabilistic reasoning in intelligent systems - networks of plausible inference,” Morgan Kaufmann, 1989.
> > >
> > > [3] J. Williamson, “Objective bayesian nets,” We Will Show Them! Essays in Honour of Dov Gabbay, Volume Two, pp. 713–730, 2005.
> > >
> > > [4] ——, “Philosophies of probability,” Philosophy of Mathematics: Handbook of the Philosophy of Science, 2009.
> > >
> > > [5] ——, “In defence of objective bayesianism,” Oxford University Press, vol. 23, no. 2, pp. 255–258, 2013.

---

> > > > ### Comment · Reviewer_gbXF · 2021-08-11
> > > > **Principles must be grounded either in experiments or in theoretically rigorous discussion**
> > > >
> > > > Principles are arbitrary assumptions (it is not a case that the references discuss fundamental theoretical concepts, not contextualised for the specific problem;), the empirical evaluation of how the AUROC/AUPR is computed to evaluate OOD detection is not (unless you propose alternative evaluation procedures), and it's based on being able to distinguish the entropies assigned to in-distribution and OOD data.
> > > >
> > > > Additionally, as I already remarked, the calibration and misclassification rejection performance are very likely harmed by underconfidence. Principles are meaningless if a classifier cannot reject a probably wrongly classified example or if it produces low confidence predictions while its accuracy is much higher than the confidence (i.e. badly calibrated). The references you provide are general philosophical references related to fundamental concepts of the bayesian theory, that are not contextualised. Debating in this hand-wavy way, one could argue that the Bernstein von Mises theorems should be wrong because it makes the posterior collapse in the asymptotic regime while one should be maximally uncertain because the maximum entropy principle says so (which is obviously wrong!). We are not doing philosophy, if you claim maximising entropy everywhere is beneficial you have to prove it with experiments over a varied set of tasks that use the distributions the network produces or produce an appropriate body of theory that in this specific context explains in detail how it works. On the contrary, well-developed theory already exists showing that maximising entropy should be done only away from in-distribution data (e.g. see Simple and Principled Uncertainty Estimation with Deterministic Deep Learning via Distance Awareness of Liu et al.)
> > > >
> > > > As for proving your method is seamless, you still haven't showed means and variances for all the evaluated metrics, so the statement is not backed by empirical evidence.

---

> > > > > ### Author Response · Authors · 2021-08-11
> > > > > **Our proposal is grounded in experiments, principles, and in rigorous discussion**
> > > > >
> > > > > We agree that the experimental evidence is fundamental.
> > > > >
> > > > > This is precisely the reason our paper has too many experimental results!
> > > > >
> > > > > ***We compared against state-of-the-art approaches and outperformed or are competitive, even operating over extremely unfair conditions.***
> > > > >
> > > > > Our method outperformed full Mahalanobis (which requires feature extraction, training additional models, adversarial examples, and hyperparameter tuning). Moreover, the proposed approach outperforms outlier exposure even without using additional data, which is indeed impressive.
> > > > >
> > > > > In this discussion, we also have a comparison with G-ODIN to show that the proposed approach produces robust results in all cases, while G-ODIN does not! Yet again, this is true even our approach operating under very unfavorable conditions.
> > > > >
> > > > > Additionally, you actually proposed to compare to an approach that we showed to have too many drawbacks to being even considered a real contest.
> > > > >
> > > > > From an entirely fair point of view, the only approach that is seamless as ours is SoftMax itself. All current OOD detection approaches present limitations and is therefore not seamless. In this fair comparison, our solution produces extremely higher performance.

---

> ### Author Response · Authors · 2021-08-13
> **Multiple seed experimental results provided with means and standard deviations for all tables and all methods**
>
> Dear reviewer,
>
> As requested, we updated all **four result tables** of our paper to include means and standard deviations of multiple seeds experiments.
>
> We appreciate the relevant feedback that helped us to improve our submission.
>
> We sincerely hope this solves your main complaint regarding our paper.
>
> https://i.imgur.com/oL4c8Nc.png
>
> https://i.imgur.com/9OhEAac.png
>
> https://i.imgur.com/QCevgIi.png
>
> https://i.imgur.com/bIPseMa.png
>
> We will **add another table** asked by another reviewer, and those values will also present means and standard deviations.

---

> > ### Author Response · Authors · 2021-08-13
> > **Brief comments regarding the new multiple seed experimental results**
> >
> > Dear reviewers,
> >
> > Please, allow us to make some brief comments regarding the new multiple seed experimental results.
> >
> > Table 2 keeps showing that SoftMax, IsoMax, and IsoMax-I produce very similar classification accuracy. Hence, IsoMax-I does not produce classification accuracy drop. We also observe that SoftMax appears to underperform when using ResNet110 on CIFAR100.
> >
> > **Hence, unlike G-ODIN, IsoMax-I present robust classification accuracy performance.**
> >
> > **Table 3 shows that IsoMax-I easily overcomes the original IsoMax for all models (x2), in-distributions (x3), out-distributions (x3), and metrics (x2) considered. Hence, in have a total of 36 (thirty-six) different situations in which IsoMax-I safely outperforms IsoMax, sometimes by a large.**
> >
> > Even being much easier to use and operating under much more restrictive conditions, Table 4 shows that IsoMax-I usually overcomes the full Mahalanobis approach or is at least competitive to it. **This was not true for the original IsoMax [1].**
> >
> > Even without requiring additional data collection, Table 5 shows that IsoMax-I always produces a similar performance of outlier exposure using either DenseNet100 or ResNet110 on CIFAR10, **which is not always true for the original IsoMax**. Even more important, using either DenseNet100 or ResNet110 on **CIFAR100**, IsoMax-I safely consistently outperforms outlier exposure, usually by a large margin! **Again, this is not always true for the original IsoMax.**
> >
> > Hence, the experiments clearly show that IsoMax-I easily overcomes outlier exposure in the fundamental few labeled data many classes setting! Additional, the former is competitive with the latter in the many labeled data few classes setting. Please, notice that outlier exposure requires the collection of extra data, while IsoMax-I does not. The additional data needed by outlier exposure increase the GPU memory requirements or doubles training time by reducing the batch size by half. Finally, we remember that outlier exposure is perfectly compatible with IsoMax-I, and hence the former may be used to improve the latter's performance.
> >
> > **The bottom line is that IsoMax-I is a major advance in terms of performance regarding the original IsoMax. The IsoMax-I improved the IsoMax to make it competitive against start-of-the-art approaches while keeping the solution still seamless.**
> >
> > [1] Macêdo, D., Ren, T. I., Zanchettin, C., Oliveira, A. L. I., and Ludermir, T. B. "Entropic out-of-distribution detection." International Joint Conference on Neural Networks (IJCNN), 2021. preprint: https://arxiv.org/abs/1908.05569.

---

> > > ### Comment · Reviewer_gbXF · 2021-08-13
> > > **Thanks for performing multi-seed experiments**
> > >
> > > I am satisfied of seeing the method presents adequate variance to be considered valuable.

---

> > > > ### Author Response · Authors · 2021-08-13
> > > > **We are working on even more experiments!**
> > > >
> > > > Dear reviewer,
> > > >
> > > > We are working on more experiments to make even more explicit in our paper that, as expected:
> > > >
> > > > **1. In-distributions get near to the prototypes, while out-distributions get far from the prototypes.**
> > > >
> > > > **2. In-distributions present higher confidence (lower entropy) than out-distributions.**
> > > >
> > > > **This shows that IsoMax-I agrees with all theoretical papers you mentioned and with which we 100% agree.**
> > > >
> > > > As we said before, the significant novelty is that IsoMax-I obtains this desired behavior using an entirely new (somewhat not intuitive and unexpected) seamless way.
> > > >
> > > > **You will also see that the original IsoMax does not get in-distributions closer to prototypes and does not make the out-distributions far from prototypes. This is why the distance score does not work for IsoMax, and it needs the entropy score to perform well, while a simple distance is an excellent score for IsoMax-I.**
> > > >
> > > > **Therefore, please note that the isomerization of the distance is a fundamental advance introduced by the IsoMax-I.**

---

> ### Author Response · Authors · 2021-08-13
> **A new paragraph was added to the conclusion's section of the paper to reflect a fair comment of the reviewer**
>
> Dear reviewer,
>
> In consequence of the reviewer's comment regarding uncertainty estimation, we added the following paragraph to the end of the conclusion section:
>
> _Finally, further studies are needed to evaluate the performance of the proposed solution regarding uncertainty estimation. Nevertheless, we are moderately optimistic considering the high OOD detection performance provided by the IsoMax-I loss and the fact that uncertainty estimation is a correlated task. Hence, we speculate that combining confidence calibration techniques with the proposed approach may produce better results than employing them in combination with the usually overconfident SoftMax loss._
>
> Thanks for contributing to improving our submission.

---

> ### Author Response · Authors · 2021-08-14
> **VERY IMPORTANT: Updating Figure 1 to make it clear that IsoMax-I is more consistent with other theoretical findings than the original IsoMax**
>
> Dear reviewer,
>
> As a consequence of our discussion regarding the importance of producing higher uncertainty/entropy for out-of-distribution and that those examples needs to be far away from the prototypes, we include histograms of uncertainty/entropy in the Figure 1.
>
> **We believe that the discussion we had greatly contributed to improve the submitted paper. We thank the reviewer for this.**
>
> We completely rewrote the legend of the mentioned figure to clarify that IsoMax-I is more consistent with other studies than IsoMax. To this, we cited the SNGP paper.
>
> Please, see the new Figure 1 in the below link:
>
> https://i.imgur.com/E5XQpNm.png
>
> **As expected and in agreement with other theoretical findings [24], BOTH ISOMAX AND ISOMAX-I PRODUCE HIGHER UNCERTAINTY ON OUT-OF-DISTRIBUTIONS.**
>
> **However, unlike original IsoMax, the **ISOMETRIZATION OF THE DISTANCES** (addition of the learnable scale, normalization of the weights, and normalization of the prototypes) made ISOMAX-I ALSO GET THE IN-DISTRIBUTION CLOSER TO PROTOTYPES AND OUT-OF-DISTRIBUTIONS FAR WAY. THIS IS NOT TRUE FOR THE ORIGINAL ISOMAX!**
>
> Therefore, this novelty introduced by IsoMax-I (**DISTANCE ISOMETRIZATION**) make it more consistent with what is recommended by other works [24] and significantly produces consistent experimental gains. Hence, IsoMax-I is both more theoretical consistent than original IsoMax and produces state-of-the-art results, while original IsoMax does not.
>
> We cited the paper [24] to make more clear in the paper that IsoMax-I is more consistent with other theoretical findings than original IsoMax. See that we cited _Simple and principled uncertainty estimation with deterministic deep learning via distance awareness_ two times in the Figure 1.
>
> ###############################################################################################
>
> So, the title of the paper is exactly the novelty:
>
> ***Improving Entropic Out-of-Distribution Detection using Isometric Distances and the Minimum Distance Score***
>
> Please, notice that the novelty is also in the abstract:
>
> _Current out-of-distribution detection approaches usually present special requirements (e.g., collecting outlier data and hyperparameter validation) and produce side effects (classification accuracy drop and slow/inefficient inferences). Recently, entropic out-of-distribution detection has been proposed as a seamless approach (i.e., a solution that avoids all the previously mentioned drawbacks). The entropic out-of-distribution detection solution comprises the IsoMax loss for training and the entropic score for out-of-distribution detection. The IsoMax loss works as a SoftMax loss drop-in replacement because swapping the SoftMax loss with the IsoMax loss requires no changes in the model's architecture or training procedures/hyperparameters. **In this paper, we propose to perform what we call an ISOMETRIZATION OF THE DISTANCES used in the IsoMax loss. Additionally, we propose to replace the entropic score with the MINIMUM DISTANCE SCORE. Our experiments showed that these simple modifications increase out-of-distribution detection performance while keeping the solution seamless.**_
>
> ###############################################################################################
>
>
> [24] Liu, J. Z., Lin, Z., Padhy, S., Tran, D., Bedrax-Weiss, T., and Lakshminarayanan, B.  Simple and principled uncertainty estimation with deterministic deep learning via distance awareness. Neural Information Processing Systems, 2020

---

### Official Review · Reviewer_M1RX · 2021-07-22

**Rating:** 5
**Confidence:** 5

**Summary:**

This paper performs isometrization of the distances in the IsoMax loss and replaces entropic score with the minimum distance score for  entropic out-of-distribution detection solution.

**Limitations And Societal Impact:**

As the overall method is simple and incremental, the novelty is limited.

**Main Review:**

The method is simple and show effective in experiments.
The authors also provide insights into why their methods work.

**Time Spent Reviewing:**

4

---

> ### Author Response · Authors · 2021-08-05
> **The Isometric IsoMax (IsoMax-I) loss is a unique out-of-distribution detection (OOD) approach that cumulatively presents an extensive set of advantages while avoiding all common drawbacks of existing approaches.**
>
> Dear reviewer,
>
> Considering that the reviewer did not make other comments or questions, we will show in this post the uniqueness of our solution compared to the existing approaches to emphasize the relevance of our work.
>
> Initially, we would like to mention that we should measure innovation in our field by the results achieved with the proposed solution and the effects/quality/correctness of the insights we added to the baseline solutions, rather than the number or the complexity of things we changed or added to it. Additionally, we should evaluate how easy our solutions are to be used in practice on a large scale by non-expert people and the potential impact in the real world of the proposed solution.
>
> **Currently, the Isometric IsoMax (IsoMax-I) loss is the _ONLY_ out-of-distribution detection (OOD) approach that _CUMULATIVELY_ presents the following characteristics (please, see also Table 1 and the introduction section of our paper):**
>
> **1. Do not present classification accuracy drop:** The experiments show that training with IsoMax-I loss does not produce classification accuracy drop. Other training-based approaches like Generalized ODIN (G-ODIN) require losing some training data for validation, consequently producing classification accuracy drop compared with training the SoftMax loss or IsoMax-I loss. Losing training data for validation is particularly harmful in the critical low labeled data regime. Classification accuracy drop is usually extremely undesired in many cases in practice.
>
> **2. Do not produce slow and energy inefficient inferences:** Approaches such as ODIN, full Mahalanobis, and G-ODIN incorporate input preprocessing, making the solution at least four times slower and energy inefficient. For real-world, large-scale adoption, we should prefer environment-friendly and low-energy consumption solutions.
>
> **3. Do not require additional data collection:** Solutions like outlier exposure require extra data collection, while the proposed approach does not. If additional data is available, outlier exposure (or similar data-driven techniques) may be used to enhance the performance of our method (see the journal version of the original IsoMax loss paper [1], Table III).
>
> **4. Do not require hyperparameter tuning:** It makes the solution readily available to be fast adopted in practice and significantly increases the overall baseline OOD performance in many areas and applications.
>
> **5. Scalability:** considering the solution consists of a SoftMax loss drop-in replacement, we have strong reasons to believe that the proposed approach scales well for large image datasets like ImageNet.
>
> **6. Domain-Agnostic:** Considering that our approach works at loss level and avoids data augmentation, it may potentially be applied to text and other types of data (see the journal version of the original IsoMax loss paper [1], Table IV).
>
> **7. Easy of use:** considering that the proposed solution works as seamless SoftMax loss drop-in replacement, using it is as simple as replacing two lines of PyTorch code.
>
> **8. Compatibility with existing inference-based approaches:** inference-based approaches (e.g., ODIN, vanilla or full Mahalanobis, outlier exposure, Gram matrices) may be applied to IsoMax-I loss pretrained model just like they are applied to SoftMax loss pretrained models to achieve even higher OOD detection performance (see the journal version of the original IsoMax loss paper [1], Table III). Unlike inference-based approaches that usually increase the computational cost of performing inferences or OOD detection, the proposed approach produces inferences as fast and computationally efficient as pure SoftMax loss trained models.
>
> **9. Competitive state-of-the-art performance even operation under more restrictive conditions and producing no side effects:** Our results show that the proposed approach overcomes ODIN despite avoiding inefficient inference. It also usually overcomes full Mahalanobis, avoiding inefficient inferences and validations using adversarial examples. Moreover, it is competitive or overcomes outlier exposure without relying on extra data. Finally, it is competitive with G-ODIN while avoiding many drawbacks (e.g., inefficient inferences, losing training data for validation, classification accuracy drop, complexity of use) even without considering all sources of overestimation presented in the experimental procedures of the G-ODIN paper (https://openreview.net/forum?id=f4jw35Vrk6d&noteId=Dg724Ga55E5, https://openreview.net/forum?id=f4jw35Vrk6d&noteId=EzOoGTMrQTg).
>
> **10. Simplicity:** The simplicity and solid foundations (e.g., distance-based loss, principle of maximum entropy, isometric distances, minimum distance score) suggest that the proposed solution generalizes well.
>
> **11. Compatibility with existing Bayesian, ensemble, and uncertainty estimation techniques:** As the IsoMax-I loss works as a SoftMax loss drop-in replacement, all available Bayesian, ensemble, and uncertainty estimation techniques may be immediately combined with it to start from a much better baseline than SoftMax loss.
>
> **12. Zero computational cost out-of-distribution detection:** Considering that the minimum distance is computed during the classification phase and that to perform out-of-distribution detection, we simply reuse this value as the score to perform OOD detection. Hence, unlike other approaches (e.g., Gram matrices), no extra computation is required.
>
> **13: Do not require feature extraction or training additional models:** After the neural network training, the solution is readily available for use. It is not necessary to perform feature extraction or to train additional models on the extracted features.
>
> Therefore, the simplicity of the solutions is actually an advantage against the other approaches rather than a drawback. Our community needs to shift from believing that a complex solution means a novelty. We should understand innovation as improving performance using the minimum amount of additional requirements/resources, producing minimum collateral side effects, and keeping the solution as simple as possible.
>
> Please, watch this: https://www.youtube.com/watch?v=KnOpWgUCtaM
> ___
> ### ***Novelty is fixing a problem without creating others!***
>
> **We need to fix what is not working without breaking what is.**
>
> **Considering that IsoMax-I loss produces much higher OOD detection performance than IsoMax loss and SoftMax loss, why should we use SoftMax loss or IsoMax loss rather than the IsoMax-I loss for training neural networks?**
>
> ***The research using IsoMax variants is at the beginning! Its simplicity gives too much space to advance even more! Accepting the paper will make us advance much fast in improving further its state-of-the-art performance. Considering that IsoMax-I is based on the principle of maximum entropy, we have strong reasons that it will generalize well.***
> ___
>
> [1] Macêdo, D., Ren, T. I., Zanchettin, C., Oliveira, A. L. I., and Ludermir, T. B. "Entropic Out-of-Distribution Detection: Seamless Detection of Unknown Examples." *Accepted for publication in IEEE Transactions on Neural Networks and Learning Systems: Special Issue on Deep Learning for Anomaly Detection.* preprint: https://arxiv.org/abs/2006.04005
>
> ___
> **Albert Einstein: _"Everything Should Be Made as Simple as Possible, But Not Simpler."_**
>
> ___
> #### ***Final remarks: Considering the reviewers did not find anything technically wrong with our approach (actually, the opposite is true, as the majority saw a robust proposal with competitive state-of-the-art performance), we suggest accepting the paper and let people decide by themselves which one (IsoMax-I loss or an existent approach) they prefer to use in practice in the real world.***

---

> ### Author Response · Authors · 2021-08-13
> **Please, notice we now have results of multiples execution of the same experiment!**
>
> Dear reviewer,
>
> Please, see the post: https://openreview.net/forum?id=f4jw35Vrk6d&noteId=FRsjpDDPKG
>
> Please, see also the comments: https://openreview.net/forum?id=f4jw35Vrk6d&noteId=Qv9bHIGJQRT

---

> ### Author Response · Authors · 2021-08-14
> **Please, notice that the title is precisely the novelty of the paper**
>
> Dear reviewer,
>
> For a discussion regarding the novelty of the paper, please see the post https://openreview.net/forum?id=f4jw35Vrk6d&noteId=U_sckqmnkp
>
> We ask you to realize that the isometrization of the distances (a learnable distance scale, the normalization of the weights, and the normalization of the prototypes) certainty contributes make IsoMax-I more theoretical consistent than the original IsoMax besides making IsoMax-I produce state-or-the-art results, which is not true for the original IsoMax.

---

> ### Author Response · Authors · 2021-08-16
> **ABOUT THE NOVELTY OF THE PAPER**
>
> Dear reviewer,
>
> We hope it is clear by now that our proposal IsoMax-I combined with the minimum distance score (MDS) is a significant advance in relation to the original IsoMax combined with the entropic score (ES).
>
> ########################
>
> Besides the **ISOMETRIZATION OF THE DISTANCES (i.e., the combination of the learnable distance scale, the normalization of the prototypes, and the normalization of the features)**, we also changed the **initialization of the prototypes** from the zero vectors used in the original IsoMax to random vectors used in IsoMax-I. Finally, we also proposed the **minimum distance score** to be used combined with IsoMax-I rather than the entropic score used with the original IsoMax.
>
> ########################
>
> By combining the modifications mentioned above, we achieved the following remarkable results:
>
> ___
>
> 1.**IsoMax-I+MDS significantly improved the OOD detection performance** in all cases (2 x models, 3 x in-data, 3 x out-data, 2 x metrics = 36 different settings) relative to the original IsoMax+ES using multi-seed experiments. Please, see the tables below:
>
> https://i.imgur.com/oL4c8Nc.png
>
> https://i.imgur.com/9OhEAac.png
>
> https://i.imgur.com/KWK7AzM.png
>
> ___
>
> 2.**Unlike the original IsoMax+ES, the proposed combination IsoMax-I+MDS produces SEAMLESS STATE-OF-THE-ART OOD detection performance** and outperforms or is at least competitive with major current approaches (e.g., full Mahalanobis, Outlier Exposure, Generalized-Odin) **even operating under much more restrictive conditions** (no hyperparameters tuning, no additional data collection, no feature extraction, no adversarial techniques and the associated problems (e.g., additional hyperparameters, slow training, classification accuracy drop)) and **producing no side effects** (no classification accuracy drop, no inefficient inferences).
>
> Again, we now have multi-seed experiments in all cases. Unlike the entropic score used by the original IsoMax, the minimum distance score used by IsoMax-I is a **zero computational cost score**, as the minimum distance is already calculated during the classification task. Therefore, the OOD detection simply reuses the same value. Please, see the tables below:
>
> https://i.imgur.com/QCevgIi.png
>
> https://i.imgur.com/bIPseMa.png
>
> ___
>
> 3.**IsoMax-I+MDS is much more consistent with recent theoretical findings** regarding what we should expect from an ideal OOD detection solution. Indeed, **ISOMAX-I FIXED A FUNDAMENTAL FLAW OF THE ORIGINAL ISOMAX.** Please, see the Figures 1c, 1d, and the correlated legend:
>
> https://i.imgur.com/E5XQpNm.png
>
> Finally, for the **motivations and theoretical intuitions that justify and explain the process of DISTANCE ISOMETRIZATION**, which we consider the main contribution of our paper and that allowed us to fix a major issue with the original IsoMax, please read this part of the submitted paper:
>
> https://i.imgur.com/TFIqdc3.png
>
> ___
>
> Finally, please also notice that all **above novelties were encapsulated into a loss, making the proposed solution _EASY OF USE_**. In other words, **only a _LOSS REPLACEMENT_ is required to use the proposed solution**.
>
> Suppose the above results generalize well for other datasets and models not contemplated in the research benchmark we use for publishing our papers (considering the simplicity of the solution, we are cautiously optimistic about this possibility). Now, imagine the *potential impact on our community* and *how easily this approach may be used* on a large scale by non-experts.
>
> ########################
>
> ***To the best of our knowledge, IsoMax-I is the only currently available seamless robust state-of-the-art performance OOD detection approach.***
>
> ########################
>
> ___
> **Albert Einstein: _"Everything Should Be Made as Simple as Possible, But Not Simpler."_**

---

### Public Comment · ~David_Macêdo1 · 2022-02-10
**Why did we decide to opt-in?**

***We decided to opt-in because we believe that the proposed approach is the first seamless (i.e., no hyperparameters tuning, no additional data collection, no classification accuracy drop, no inefficient inferences) out-of-distribution detection solution that produces competitive state-of-the-art performance*.***

Additionally, besides avoiding all current approaches limitations (e.g., Mahalanobis, Outlier Exposure, Generalized ODIN) and producing very high out-of-distribution detection performance, the proposed solution may be fast and easily used by simply changing the loss function utilized to train the model, making it readily available to be applied on a large scale also by non-experts in the field. Furthermore, the out-of-distribution detection is performed using a zero computational cost score.

Moreover, extensive experiments were provided using the de fact standard out-of-distribution detection benchmark proposed by Dan Hendrycks and Kevin Gimpel (https://arxiv.org/abs/1610.02136). The mentioned baseline includes many datasets, models, and metrics. Each type of experiment was executed many times and the mean and standard deviations were provided. The paper has 5 (five) tables with plenty of experimental results, and they are very compelling regarding the effectiveness of the proposed modifications. We additionally presented intuitions and analyses of why the solution works. We believe that the paper is clearly written. Moreover, considering the simplicity of the solution and that simple approaches usually scale and generalize better than complex ones, we are cautiously optimistic that the presented solution may work well also on different datasets and models.

**Despite "simple" (which we consider a remarkable advantage rather than a drawback), all modifications are significant, as the 5 (five) changes proposed were simultaneously necessary to make the overall solution work properly. If a single one of these alterations is removed, the overall solution clearly fails to provide competitive state-of-the-art performance.**

Like the original IsoMax from 2019 (https://arxiv.org/pdf/1908.05569v1.pdf), the proposed solution is also a distance-based loss, which is the type of approach that the community is recently slowly converging to believe it is the way to follow to tackle out-of-distribution detection. Unlike the original IsoMax from 2019 (the first distance-based loss used to directly training the neural network without feature extraction aiming to tackle out-of-distribution detection), the proposed modification presents high out-of-distribution detection performance using both the entropy and the minimum distance as a score. The novel version of IsoMax easily outperforms the original one.

Finally, we also imagine that the discussion on this page may be somehow useful to people interested in out-of-distribution detection.

For the updated version of the paper after the contributions of the reviewers, please see:

https://arxiv.org/abs/2105.14399

Please, use the above arXiv version or preferentially a more recent one possibly published in another venue if you desire to cite our paper.

For the updated version of the code that allows using the proposed solution only changing two lines of code, please see:

https://github.com/dlmacedo/entropic-out-of-distribution-detection

Naturally, the above code may also be used to reproduce our results.

Developing easy-to-use approaches is the way to go to bring effective solutions to a broader audience. This is why we insist on such approaches. There is no rational reason to use complex solutions when a simple one performs similar or better, besides avoiding all severe drawbacks of the complex proposals.

**We believe that neural networks may be made much more reliable in seamlessly detecting unknown examples by simply replacing the loss used to train them. Hence, we think this is a relevant research path to build a more reliable AI.**

Albert Einstein: "Everything Should Be Made as Simple as Possible, But Not Simpler."
___

*Generalized ODIN (G-ODIN) produces relevant classification accuracy drop in some cases (e.g., ResNet34 trained on CIFAR100). The authors recommend retraining the model using an apparent fine-tuned regularization (i.e., dropout with a value of p=0.7) to mitigate this problem. After that, the classification accuracy drop persists, but it becomes less intense. However, besides the inconvenience of training the network once more, the mentioned procedure makes the out-of-distribution detection performance plunge significantly. Therefore, the proposed fix does not work properly.

Moreover, considering that CIFAR100 has 500 training examples per class, we believe this overfitting problem may become even worse when dealing with datasets with fewer examples per class, which are critically important and largely used in practice (e.g., medical image datasets, health applications). The overfitting may be explained by the fact that G-ODIN adds an entire fully connected layer to the end of the neural network, while our solution adds a single parameter.

Furthermore, except for the addition of inefficient inference input preprocessing, G-ODIN is very similar to the previously published Scaled Cosine approach, whose authors openly admit that the mentioned solution produces relevant classification accuracy drop.

G-ODIN has many flavors, making it hard to know beforehand which one will perform better in practice without training the same network many times using the several distinct options. After that, performing model selection is hard, as we are not supposed to have access to out-of-distribution data to validate the best model.

Summarily, considering the G-ODIN produces classification accuracy drop in some cases, we do not think it as a seamless out-of-distribution detection approach. Lastly, it is not as easy to implement and to use as our approach.

---

### Decision · Program_Chairs · 2021-09-27

**Decision:**

Reject

**Comment:**

The extensive discussion here seems to have resulted in some substantial improvements to your paper, but there remain some concerns about the thoroughness of the experimental results (in particular with respect to the choice of architecture and of datasets) and clarity of the discussion. Some reviewers complain that the changes in your proposal are too simple; I want to emphasize that I do _not_ think this is a good reason to reject a paper (and indeed have fought to accept "too simple" papers in the past), but that "simple" methods do tend to come with a higher bar for clarity and thoroughness of the presentation. The changes brought up in this discussion were also of somewhat significant scope. Thus, it seems best if you integrate the results of the discussion here into a revision to be reviewed at another venue.